# ImpMIA: Leveraging Implicit Bias for Membership Inference Attack

## Abstract

Determining which data samples were used to train a model, known as Membership Inference Attack (MIA), is a well-studied and important problem with implications on data privacy. SotA methods (which are black-box attacks) rely on training many auxiliary reference models to imitate the behavior of the attacked model. As such, they rely on assumptions which rarely hold in real-world settings: (i) the attacker knows the training hyperparameters; (ii) all available non-training samples come from the same distribution as the training data; and (iii) the fraction of training data in the evaluation set is known. We show that removing these assumptions significantly harms the performance of black-box attacks. We introduce **_ImpMIA_**, a Membership Inference Attack that exploits the **_Implicit Bias_** of neural networks. Building on the maximum-margin implicit bias theory, _ImpMIA_ uses the Karush–Kuhn–Tucker (KKT) optimality conditions to identify training samples – those whose gradients most strongly reconstruct the trained model's parameters. Our approach is optimization-based, and requires NO training of reference-models, thus removing the need for any knowledge/assumptions regarding the attacked model's training procedure. While _ImpMIA_ is a white-box attack (a setting which assumes access to model weights), this is becoming increasingly realistic given that many models are publicly available (e.g., via Hugging Face). _ImpMIA_ achieves SotA performance compared to both black and white box attacks in settings where only the model weights are known, and a _candidate set_ containing some of the training data is available.

## 1 Introduction

Ensuring that trained models do not leak information about their training sets is a critical challenge. Membership inference attacks (MIAs) evaluate this risk by determining whether a given example was part of a model's training data. MIAs can be broadly divided into two categories: black-box, which assume only query access to model outputs (Shokri et al., 2017; Yeom et al., 2018; Li & Zhang, 2021; Carlini et al., 2022; Wu et al., 2024; Peng et al., 2024), and white-box, which exploit access to internal parameters such as weights or gradients (Nasr et al., 2019; Leino & Fredrikson, 2020; Cohen & Giryes, 2024; Suri et al., 2024).

While access to model parameters is a restrictive assumption, this scenario is increasingly realistic, as many modern models are released with their parameters publicly available (e.g., via platforms such as Hugging Face). Existing white-box MIAs exploit this access by using internal quantities such as gradients and activations (Nasr et al., 2019; Leino & Fredrikson, 2020), or influence scores (Cohen & Giryes, 2024). These methods typically assign membership scores to candidates individually, based on local per-sample signals that capture how the model behaves on each candidate. Although such signals can distinguish members from non-members on average, individual non-members may also exhibit strongly member-like behavior. This is particularly problematic under the stringent and reliable evaluation criterion suggested by Carlini et al. (2022), where even a small number of highly scored non-members can substantially reduce performance. One possible reason is that these methods rely primarily on local per-sample signals, effectively asking whether the model behaves as though it has seen a given sample. A more informative global use of the model parameters may be to jointly identify which candidate samples are most important for explaining the learned parameters, providing a signal of their likelihood of being training members.

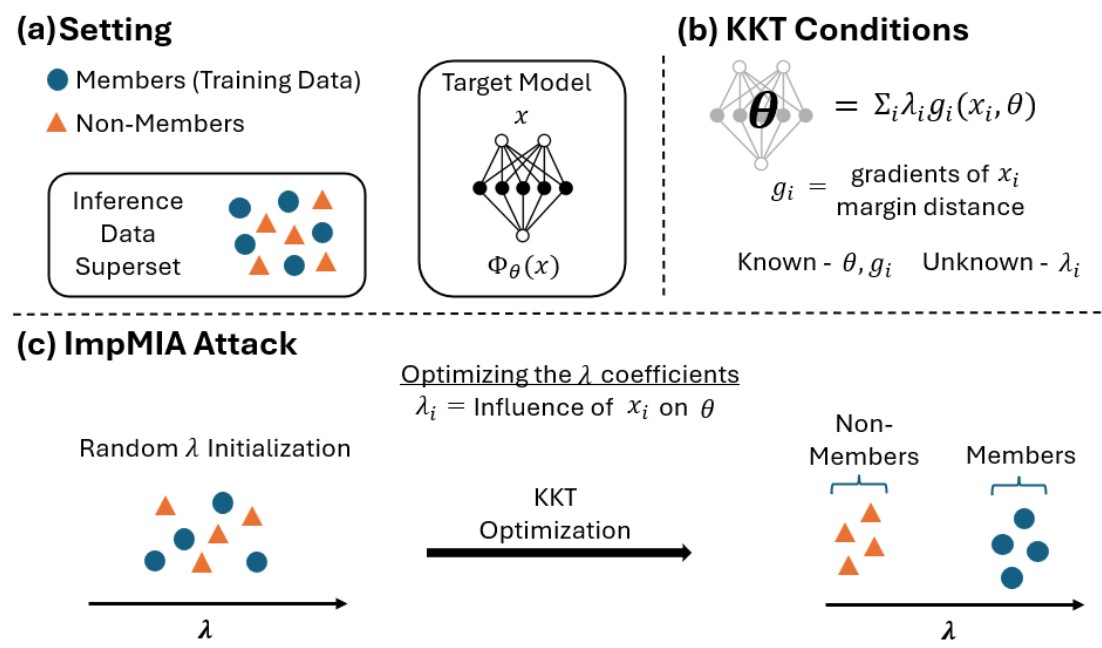

Figure 1: **Overview of the Approach.** **(a)** Setting: Given a trained model with parameters $\theta$ and a candidate set containing training *members* (blue) and *non-members* (orange), the adversary's goal is to identify which samples are members. **(b)** KKT conditions: Our attack builds on implicit bias theory, which shows that gradient-based optimization converges to solutions satisfying the Karush–Kuhn–Tucker (KKT) conditions of the maximum-margin problem. Since weights are known and gradients are computable, only the coefficients remain unknown. **(c)** ImpMIA: We optimize one coefficient per sample to best reconstruct the model parameters, where members are expected to receive large coefficients and non-members small ones.

Separately from the level of access to the target model, MIAs differ in the amount of auxiliary knowledge they require. In particular, many leading black-box attacks rely on reference models and therefore assume substantial information about the target model and its training setting. These methods estimate the loss distributions of members and non-members by training auxiliary models intended to mimic the target model's behavior. However, training large sets of reference models is computationally expensive, and—more importantly—their effectiveness depends on the reference models being accurate surrogates of the target. As a result, these attacks rely on several strong assumptions: (i) the attacker knows the training hyperparameters (e.g., learning rate, optimizer, number of epochs); (ii) the non-training samples come from the same distribution as the training data; and (iii) the fraction of training members in the evaluation set is known. When any of these assumptions is violated, the performance of black-box MIAs drops significantly (Shokri et al., 2017; Salem et al., 2019; Song & Mittal, 2021; Carlini et al., 2022), limiting their reliability for auditing privacy.

We propose *ImpMIA*, a white-box membership inference attack that introduces a new global signal inspired by neural network *implicit-bias* theory (see the approach overview in Fig. 1). Rather than deriving a separate score for each candidate, *ImpMIA* obtains all membership scores jointly through a shared optimization that seeks to explain the learned model parameters. At the same time, unlike reference-model-based attacks, it does not require training auxiliary models or knowledge of the target model's training hyperparameters, data distribution, or member ratio. Thus, *ImpMIA* leverages white-box access to the model parameters while operating in a no-auxiliary-knowledge setting, requiring only the trained model weights and a candidate set that contains the training data.

Our attack builds on the implicit-bias theory in neural nets, which shows that gradient-based optimization tends to converge to solutions that satisfy the Karush–Kuhn–Tucker (KKT) optimality conditions of a certain maximum-margin problem (Lyu & Li, 2019; Ji & Telgarsky, 2020). In practice, this implies that the trained

parameters of a network can be approximately expressed as a linear combination of per-sample gradients from the training set. Given a set of candidate samples and the trained network weights, we *optimize* a set of coefficients—one per sample—that best reconstructs the network parameters. This provides the key signal: training samples are expected to receive significantly larger coefficients, while non-members remain small.

*ImpMIA* is best suited for scenarios where the training-set is contained in the examined-set. Such scenarios are relevant, e.g., if one wants to check whether their own dataset has been used for training a model, or when a dataset is *known* to contain the training-set (e.g., a publicly available dataset), but not which part of it was used for training. While this is a somewhat restrictive assumption, we show that it can be relaxed (App. B.3), and that our performance is competitive even at low coverage of the training-set (only 10%), and becomes superior at high coverage.

*ImpMIA* achieves superior results, surpassing both black-box and white-box attacks under no auxiliary-knowledge settings across 3 benchmark datasets. Our extensive analysis shows that removing the knowledge assumptions made by most reference-model-based methods (knowledge of the attacked model's training procedure & setting), leads to a significant drop in performance of SotA methods, while our attack remains unaffected. Altogether, these results highlight the importance and effectiveness of our proposed method as a practical membership inference attack.

**Our contributions are as follows:**

- We introduce ***ImpMIA***, the first membership inference attack based on the implicit bias of gradient descent and its corresponding KKT conditions.

- ***ImpMIA*** achieves SotA performance in scenarios where only model weights & a candidate data pool are available.

- We provide a systematic evaluation of the robustness of leading MIA methods in settings where training hyperparameters, data distribution, or member ratios are unknown.

## 2 Related Work

### 2.1 Membership inference

Membership Inference Attacks (MIAs) are divided into black-box attacks, which rely on model outputs, and white-box attacks, which exploit access to model parameters.

**White-box** attacks exploit access to a model's internal parameters (often gradients) to amplify membership signals. Nasr et al. (2019) introduced one of the first frameworks, leveraging activations and per-example gradients. Sablayrolles et al. (2019) derived the Bayes-optimal test under white-box access, showing that maximum membership power can be achieved by computing likelihood ratios over model parameters. Leino & Fredrikson (2020) showed in their Stolen Memories attack that gradient norms alone provide strong membership signals, and that training an auxiliary classifier to distinguish gradients from members versus non-members further improves accuracy. Most recently, Cohen & Giryes (2024) proposed a self-influence attack that uses influence functions to measure each sample's effect on its own loss, combined with the predicted label. While white-box access is a strong assumption, it is increasingly realistic as many modern models are released with their full weights (e.g., on Hugging Face).

**Black-box MIAs** assume the attacker can only query the target model and observe its outputs. Shokri et al. (2017) introduced the shadow-model framework, training reference models to mimic the target and then learning an attack model from their outputs. Yeom et al. (2018) later showed that even without reference models, the simple "gap" heuristic, predicting membership when the model's output label matches the ground truth, can be effective. Li & Zhang (2021) proposed decision-based attacks relying on adversarial perturbations, while Ye et al. (2022) introduced *Attack-P*, a population-based loss thresholding method, and *Attack-R*, a sample-specific calibration using percentiles from reference models for improved robustness. Building on the reference-model paradigm, Carlini et al. (2022) proposed LiRA, which compares target losses to reference-model distributions and emphasized low false-positive evaluation. Zarifzadeh et al. (2023) (RMIA)

further refined LiRA with an optimized likelihood-ratio test, improving efficiency under strict computational limits. LiRA and RMIA currently represent the strongest-performing black-box MIAs.

**Limitations of Reference-Model Attacks.** Reference-model attacks are costly to train and their effectiveness depends heavily on how these models are trained. Specifically, three key assumptions; whose violation significantly reduces MIA performance: (i) ***Knowledge of the target model's training hyperparameters*** (e.g., learning rate, optimizer, epochs): Jayaraman et al. (2021) and Carlini et al. (2022) (LiRA) reported accuracy drops when altering those hyperparameters, while Pradhan et al. (2025) propose methods to partially mitigate this hyperparameters sensitivity, but rely on additional reference models and distributional assumptions. (ii) ***Matching data distribution***: Salem et al. (2019) and Shi et al. (2024) show degraded accuracy when shadows were trained on different domains (e.g., CIFAR-10 for a CIFAR-100 target). (iii) ***Member ratio in the inference pool***: Jayaraman et al. (2021) and Song & Mittal (2021) found inflated false positives and reduced accuracy when this assumption was wrong. In this work, we systematically test these factors showing significant drops across models on CIFAR-10, CIFAR-100, and CINIC-10. Such scenarios where these factors are *unknown* to the attacker are common as models, particularly those trained on sensitive data, are unlikely to publish this information.

## 2.2 Implicit Bias of Gradient Descent

In overparameterized neural networks, one might expect overfitting the training data. Yet gradient-based methods tend to converge to classifiers that generalize well to new unseen data (Zhang et al., 2021; Neyshabur et al., 2017). This phenomenon is explained by the *implicit bias* of training algorithms: gradient descent tends to prefer specific solutions, and characterizing these has been central to deep-learning theory in recent years (see Vardi (2023) for a survey). For homogeneous ReLU networks trained to zero logistic or cross-entropy loss, Lyu & Li (2019) and Ji & Telgarsky (2020) showed that the learned weights necessarily satisfy the *KKT conditions* of a maximum-margin problem. Building on this, Haim et al. (2022) demonstrated that networks trained with binary cross-entropy allow reconstruction of dozens of nearly pixel-perfect training samples. This was later extended to multiclass classifiers (Buzaglo et al., 2023) and more realistic transfer-learning workflows (Oz et al., 2024). All of these data reconstruction attacks are limited to very small datasets (up to a few thousand examples), and require simple models (mostly MLPs). In this work, we adapt the implicit-bias approach for the first time to membership inference attack, identifying which samples from a candidate pool best satisfy the KKT conditions. Finally, in a recent theoretical work, Smorodinsky et al. (2024) studied when the implicit bias result of Lyu & Li (2019) provably leads to privacy vulnerabilities in simplified settings.

## 3 Preliminaries

### 3.1 Membership Inference Attack Setting

Membership inference attacks (MIAs) aim to determine which data points were part of the training set of a machine learning model. The strongest recent black-box methods are reference-model based, and they rely on additional assumptions about the target model's training configuration and the candidate set, which are unlikely to hold in practice. In this work, we focus on a scenario where the adversary has access only to a candidate set that contains the training set (or part of the training set) and to the model weights. This reflects real-world conditions: modern models are often released publicly with their weights, while auditors may possess large candidate pools that include the training set but lack detailed knowledge of the training data distribution or the exact training configuration. Our setting adapts the basic membership inference game (Yeom et al., 2018; Jayaraman et al., 2021), where a single sample is evaluated at a time, into a *set-based* formulation in which the full candidate pool is attacked and evaluated (similar to the online setting in Carlini et al. (2022); Zarifzadeh et al. (2023)). We assume that the evaluated set contains the training set or at least a portion of it (e.g. more than 10%). While this assumption was not taken in prior work, the reported results in those settings were in fact obtained under it. That is, for technical reasons, their evaluations were obtained in a setting where the attacker uses a candidate set containing the training data (Carlini et al., 2022). Results for our method in a scenario where this assumption is relaxed are presented in App. B.3.

Formally, we study a *set-based* No-Auxiliary-Knowledge setting. Let $X_{\text{train}}$ be the (unknown) training set drawn from a distribution $\mathcal{D}$, and $f_\theta$ a model trained on $X_{\text{train}}$. Let $X_{\text{sup}}$ be a candidate pool with $X_{\text{train}} \subseteq X_{\text{sup}}$, and the remaining samples being non-members, potentially drawn from other distributions. The adversary is given the trained parameters $\theta$ and the pool $X_{\text{sup}}$, but: (i) does not know the target model's training hyperparameters; (ii) cannot assume non-members are drawn from the same distribution as the training set; and (iii) does not know how many members are in $X_{\text{sup}}$ or their ratio. The adversary must then assign a real-valued score to each sample, with membership decisions.

### 3.2   The Implicit Bias Formulation

In this section, we provide an overview of the KKT conditions and the maximum-margin formulation, following the definitions in Haim et al. (2022); Buzaglo et al. (2023). While the theory described below is formally constrained to homogeneous[1] ReLU networks (Lyu & Li, 2019; Ji & Telgarsky, 2020), we show in practice that the results hold more generally for other architectures. The theoretical results of Lyu & Li (2019); Ji & Telgarsky (2020) consider training without weight decay, but in App. D we show that incorporating weight decay leads to the same final equations, and in App. B.9 we analyze its influence on the attack performance.

***Implicit bias of gradient flow on homogeneous networks***: Let $\Phi(\theta; \cdot) : \mathbb{R}^d \to \mathbb{R}^C$ be a homogeneous ReLU network with logits $\Phi(\theta; x)$. Consider minimizing the cross-entropy loss over a multiclass dataset $\{(x_i, y_i)\}_{i=1}^n \subseteq \mathbb{R}^d \times [C]$ using gradient flow (i.e., gradient descent with infinitesimally small step size). Suppose that at some time $t_0$ the network classifies all training samples correctly. Then gradient flow converges in direction to a KKT point of the multiclass maximum-margin problem:

$$\min_\theta \tfrac{1}{2}\|\theta\|^2 \quad \text{s.t.} \quad \Phi_{y_i}(\theta; x_i) - \Phi_j(\theta; x_i) \geq 1, \quad \forall i \in [n], \ \forall j \in [C] \setminus \{y_i\}.$$

The corresponding KKT conditions for solving this maximum margin problem are:

$$\theta - \sum_{i \in [n]} \sum_{j \in [C] \setminus \{y_i\}} \lambda_{i,j} \, \nabla_\theta \big[ \Phi_{y_i}(\theta; x_i) - \Phi_j(\theta; x_i) \big] = 0, \tag{1}$$

$$\forall\, i \in [n], \ j \in [C] \setminus \{y_i\} : \ \begin{cases} \Phi_{y_i}(\theta; x_i) - \Phi_j(\theta; x_i) \ \geq \ 1, \\ \lambda_{i,j} \ \geq \ 0, \\ \lambda_{i,j} = 0 \ \text{if } \Phi_{y_i}(\theta; x_i) - \Phi_j(\theta; x_i) \neq 1. \end{cases} \tag{2}$$

Equation 1, called the *stationarity condition*, represents the weights as a linear combination of margin gradients (distance between a sample's true class $\Phi_{y_i}$ and other classes $\Phi_j$), while Equation 2 specifies additional constraints. The coefficients are the $\lambda_{i,j}$, defined per sample i per class j. In practice, the distance of a sample $x_i$ to the decision boundary is typically determined by a single competing class $j$. Following Buzaglo et al. (2023), we therefore simplify the stationarity condition by retaining only the smallest margin:

$$\theta \ = \ \sum_{i=1}^m \lambda_i \, g_i, \quad\quad g_i = \nabla_\theta \Big[ \Phi_{y_i}(x_i; \theta) - \max_{j \neq y_i} \Phi_j(x_i; \theta) \Big]. \tag{3}$$

While our attack builds on the same condition as Buzaglo et al. (2023), our goal differs: instead of reconstructing training data $\{x_i\}$, we fix the candidate inputs and optimize only the coefficients $\{\lambda_i\}$ to obtain membership

## 4   ImpMIA Attack

In this section, we introduce *ImpMIA*, our white-box membership inference attack that exploits the implicit bias of neural networks. The attack builds on the observation from Eq. 3 that trained parameters can be

---

[1]A model $\Phi$ is homogeneous w.r.t. $\theta$ if there exists $L > 0$ such that $\forall c > 0, x : \ \Phi(x; c\theta) = c^L \Phi(x; \theta)$.

represented as a linear combination of per-sample gradients from the training samples (members). Thus, members can be distinguished from non-members by their relative contribution to this representation of the parameters, where members contribute to this reconstruction while non-members do not. While this theory motivates our approach, the practical attack is not theoretically guaranteed, since the architectures and training procedures used in our experiments do not exactly satisfy all of the formal assumptions. We first outline the practical construction of the attack in Section 4.1, and then detail the technical optimization procedures and stabilization strategies in Section 4.2.

### 4.1 Practical Attack Construction

Building on the theoretical link between the KKT stationarity condition and the representation of trained parameters (Eq. 1), we now describe how this insight is used to devise the ImpMIA attack. The KKT stationarity condition guarantees that the trained parameter vector can be expressed as:

$$\theta = \sum_{i \in X_{\text{train}}} \lambda_i \, g_i,$$

where each $g_i$ is the margin gradient of a training sample and $\lambda_i \geq 0$ is its corresponding multiplier. Equivalently, the KKT stationarity condition implies that the trained parameter vector lies in the subspace spanned by the margin gradients of the training samples. From this perspective, ImpMIA searches over the candidate pool for the samples whose gradients best span, or reconstruct, the learned parameter vector. The model weights are known hence the per-sample gradients can be computed. Therefore, the only unknowns are the $\lambda$ coefficients, which can be obtained by optimizing them to satisfy the equation.

In practice, the attacker does not know the true training set, but only has access to a candidate pool $X_{\text{sup}} = \{(x_i, y_i)\}_{i=1}^{M}$, which contains an unknown subset of training samples $X_{\text{train}}$ and non-members $X_{\text{test}}$. However, we can still optimize the coefficients using all samples, deriving a $\lambda$ coefficient for each (either member or not). This provides the key signal: we expect the coefficients of training samples to be significantly larger, while those of non-members remain small. This is because the number of network parameters is typically much larger than the size of the candidate pool, and therefore deriving the correct weights in the optimization is much more likely when true members exert stronger influence. Importantly, when $|X_{\text{sup}}|$ is smaller than the dimension of $\theta$ and the vectors $\{g_i\}$ are linearly independent, the system admits a unique solution for $\{\lambda_i\}$. Note that, following Eqs. (2)-(3), we expect large coefficients for training samples near the margin (close to the decision boundary), while other samples are expected to have lower coefficients (zero in theory).

Formally, for each candidate $(x_i, y_i)$ we compute the multiclass margin gradient

$$g_i = \nabla_\theta \Big[ \Phi_{y_i}(\theta; x_i) - \max_{j \neq y_i} \Phi_j(\theta; x_i) \Big],$$

and stack these into the matrix $A = [g_1 \mid \cdots \mid g_M] \in \mathbb{R}^{p \times M}$,

where $p$ is the number of model parameters. If the training set were known, we could restrict to $A_{\text{train}}$ and solve exactly for multipliers $\lambda_{\text{train}}$ such that $A_{\text{train}} \lambda_{\text{train}} = \theta$. Since the training set is unknown, we instead solve the full system $A \lambda_{\text{score}} = \theta$, deriving $M$ (number of candidates) coefficients. The resulting coefficients are used to calculate a membership score, where large values for a specific sample are interpreted as evidence of a higher probability that this sample was part of the training data (i.e., a member). To improve robustness and suppress spurious large values from non-members, we incorporate additional techniques for regularization and aggregation (detailed in Section 4.2). In Fig. 2, we present a scatter plot of members and non-members, where the y-axis shows the $\lambda$ score and the x-axis the sample's distance from the margin (i.e., $\Phi_{y_i}(x_i; \theta) - \max_{j \neq y_i} \Phi_j(x_i; \theta)$). As illustrated, high $\lambda$ scores are strong indicators of membership.

## 4.2 Implementation Details

Given an evaluation set of labeled samples, we optimize the $\lambda$ coefficients to satisfy the KKT conditions. The overall pipeline includes the following parts: (i) **_pre-filtering_** of candidate samples based on their classification margin, (ii) **_augmentation_** using horizontal flips, (iii) **_block division_** of weights for more efficient optimization, (iv) **_gradient matrix construction_** per block-structured gradient matrix $A$ over the chosen model parameters, (v) **_optimization_** to solve the KKT conditions and learn the coefficients $\lambda$ via blockwise optimization with dedicated regularization, (vi) **_coefficient aggregation_** across blocks into robust per-sample scores, and finally (vii) **_post-processing_** of the coefficient scores to derive the final per-sample score. See Algorithm 1 for a pseudo code of the pipeline.

We first filter out misclassified samples, since training members are more likely to be correctly predicted. For each remaining sample, we include both the original and its horizontal flip, reflecting the fact that most models are trained with augmentations, while keeping computational cost low by not adding further augmentations.

The only optimized variables are the $\lambda$ coefficients. Therefore, we precompute the margin gradients with respect to the model parameters, forming the columns of the matrix $A$ (as introduced in the previous section). Since $A$ is extremely large (with the number of rows corresponding to the number of network parameters), we split it into blocks for more efficient optimization. Specifically, each block corresponds to roughly $1.5 \times 10^5$ parameters, and we optimize the coefficients separately per block. This block partitioning reduces memory usage and improves optimization stability. In our implementation, blocks are constructed sequentially, so parameters from the same or nearby layers/filters are grouped together. This avoids mixing parameters with very different gradient scales and statistics in the same optimization problem. Moreover, although non-members can obtain a high coefficient due to imperfect optimization, deriving coefficient solutions per block allows averaging across blocks. This improves results, since a sample is unlikely to consistently receive high coefficients in different blocks if it was not part of the training set.

Both the gradient block and the target parameter vector are centered and normalized. For each sample, we then aggregate the optimized coefficients across blocks and augmentations to obtain a final score (see App. A for more details). Our attack is designed to be both memory-efficient and numerically stable, while suppressing noisy scores for non-members.

# 5 Experimental Setting

We evaluate **ImpMIA** against both black-box and white-box baselines across CIFAR-10, CIFAR-100, and CINIC-10, using a standard ResNet-18 as the target model. Our experiments reflect realistic adversarial conditions, where the attacker is given only the trained model weights and a candidate set which includes the training data, with no knowledge of the training hyperparameters, data distribution, or member ratio. We first describe the datasets, evaluation metrics, and the competing baseline models (Section 5.1). We then detail the different scenarios corresponding to the assumptions usually made by reference-model-based MIAs, beginning with the setting where all assumptions are provided, and then moving to a more realistic setting without them (Section 5.2).

## 5.1 Models, Datasets and Evaluation Metrics

For all experiments, the target model is a ResNet-18 trained following the standard recipe of Cohen & Giryes (2024). Specifically, we use a batch size of 100, a learning rate of 0.1, momentum 0.9 with Nesterov acceleration, weight decay of $10^{-4}$, and train for 400 epochs using stochastic gradient descent (SGD) with standard data augmentations (random crop and horizontal flip). For each dataset and scenario, we randomly sampled 5 different sets of training samples, and trained 5 target models. Results are averaged across them.

We report performance using both aggregate metrics (AUC) and stringent low-false-positive criterion, True Positive Rate (TPR) at False Positive Rate (FPR) of 0.01% and 0.0%. This criterion was introduced in Carlini et al. (2022), which observed that average-case metrics such as accuracy or AUC can be misleading: an attack may appear strong overall, yet fail completely in the regime of low false positives, which is the

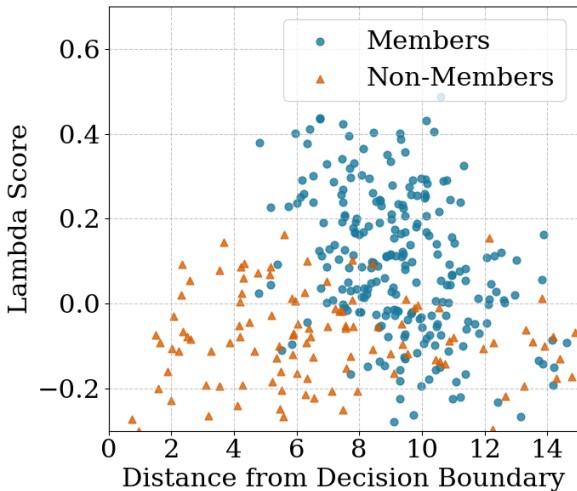

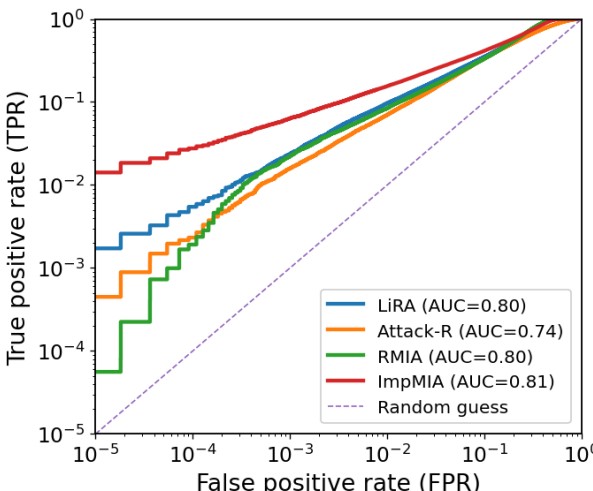

Figure 2: **Lambda scores visualization.** Scatter plot of evaluated samples, with the x-axis showing distance to the decision boundary and the y-axis showing $\lambda$ scores; points are colored by membership (member vs. non-member). High $\lambda$ indicates membership.

Figure 3: **TPR-FPR in the No-Auxiliary-Knowledge setting (CIFAR-10).** We evaluate a setting with (i) unknown training configuration, (ii) distribution shift in the candidate pool, and (iii) unknown member ratio.

regime most relevant for privacy auditing (see App. E for a detailed discussion). In practice, an adversary cannot afford a large number of false alarms, as even a tiny FPR may translate to thousands of incorrectly flagged samples in real-world deployments. Therefore, evaluating TPR at very low FPR provides a stricter and more meaningful measure of membership inference risk.

We compare ImpMIA against recent state-of-the-art black-box and white-box membership inference attacks. For black-box attacks, we include Attack-P (Li & Zhang, 2021), Attack-R (Ye et al., 2022), and focused on the online version of LiRA (Carlini et al., 2022) and RMIA (Zarifzadeh et al., 2023), which currently represent the strongest black-box MIAs. For black-box baselines, we followed their best setting (e.g. training 256 reference models per experiment). For white-box attacks, we evaluate the adaptive self-influence attack (AdaSIF) (Cohen & Giryes, 2024), which represents the current state of the art in this category (see App. C.2 for additional implementation details). We also include as a baseline a simple white-box attack based on the magnitude of the network gradients, since Nasr et al. (2019) noted that this quantity provides the main signal in their attack (see App. C.2 for additional details). We were unable to compare with Nasr et al. (2019) and Leino & Fredrikson (2020), as their code is not available.

## 5.2 Membership Inference Attack Scenarios

The basic scenario is the one commonly assumed in black-box attacks, which remain the strongest methods even relative to existing white-box approaches. In the *Full Auxiliary-Knowledge setting*, the attacker is assumed to know both the model architecture and the training hyperparameters used to train the target model (e.g., learning rate, optimizer, epochs). The attacker is also given a pool of samples containing both members and non-members, under the assumptions that (i) the pool is drawn from the same distribution as the training data and (ii) the member/non-member ratio is known, often fixed at 1:1.

These assumptions give reference-model attacks (e.g., LiRA, RMIA) a strong advantage: by training many reference models under identical conditions, they can reconstruct the target model's member/non-member loss distributions. This explains their strong performance in this regime, though it comes at the cost of training large model ensembles and relies on knowledge rarely available in practice.

| Attack | CIFAR-10 | | | CIFAR-100 | | | CINIC-10 | | |
|---|---|---|---|---|---|---|---|---|---|
| | AUC | @0.01 % | @0.0 % | AUC | @0.01 % | @0.0 % | AUC | @0.01 % | @0.0 % |
| *Attack-P* | 0.76 | 0.02 | 0.0 | 0.89 | 0.01 | 0.0 | 0.83 | 0.01 | 0.0 |
| *Attack-R* | 0.74 | 0.23 | 0.04 | 0.95 | 0.52 | 0.04 | 0.83 | 0.31 | 0.0 |
| *LiRA* | 0.80 | 0.55 | 0.17 | 0.96 | 7.90 | 2.36 | **0.88** | 2.27 | 0.66 |
| *RMIA* | 0.80 | 0.19 | 0.01 | **0.97** | 6.73 | 1.22 | 0.87 | 0.15 | 0.03 |
| *GradNorm – loss* | **0.81** | 0.11 | 0.01 | 0.93 | 0.10 | 0.04 | 0.85 | 0.09 | 0.01 |
| *GradNorm – margin* | 0.72 | 0.02 | 0.0 | 0.81 | 0.02 | 0.01 | 0.77 | 0.03 | 0.01 |
| *AdaSIF* | 0.80 | 0.05 | 0.0 | 0.92 | 0.01 | 0.0 | 0.85 | 0.01 | 0.0 |
| ***Ours*** | **0.81** | **2.76** | **1.41** | 0.95 | **14.86** | **5.26** | 0.87 | **5.32** | **2.47** |

Table 1: **Membership Inference results.** Performance of ImpMIA compared to black-box (LiRA, RMIA) and white-box baselines across 3 datasets under the No-Auxiliary-Knowledge setting. The main relevant metrics are TPR values at fixed false-positive rates (FPR = 0.0% and 0.01%), which capture detection power under stringent error constraints. ImpMIA significantly surpasses all other methods by a wide margin, due to their reliance on the different assumptions. For completeness, we also report AUC as an aggregate measure.

To evaluate attack performance under more realistic conditions, we design experimental scenarios that relax these assumptions. In Section 6, we present results in the more realistic setting where those assumptions are not made, as well as in each individual scenario where one assumption is removed. Importantly, to ensure fair comparison across scenarios and with prior work, we kept the target model fixed and followed the basic scenario used in previous studies.

**The assumption-elimination scenarios are:**

- ***Unknown Training Configuration*** – Since the attacker is not exposed to the target model's training parameters, we trained the reference models of methods that require them using different settings. Specifically, we used a different batch size (200 instead of 100), learning rate (0.01 instead of 0.1), weight decay ($10^{-3}$ instead of $10^{-4}$), and epochs (100 instead of 400) than those used for the ResNet-18 target model (detailed in Section 5.1).

- ***Different Data Distribution*** – In this case, the attacker's candidate pool mixes in-distribution data (the distribution from which the target model was trained) and out-of-distribution (OOD) data. For each dataset, we construct a pool of $50k$ samples by combining $30k$ in-distribution images with $20k$ OOD images (taken from another dataset). The target model is trained on $25k$ in-distribution samples drawn from the $30k$ portion, allowing for 5 repetitions of the experiment (with 5 different target models). In the reference-based attacks under the relevant online setting, the attacker trains each reference model on half of the evaluation set, i.e., $25k$ examples sampled from the full mixed pool. OOD sources include a subset of ImageNet adapted to CIFAR-10 (CINIC-10 (Darlow et al., 2018)) and an enriched OpenImages dataset (Kuznetsova et al., 2020). Specifically, CINIC-10 is used as the OOD source for CIFAR-10, while OpenImages is used as the OOD source when CINIC-10 is the in-distribution dataset.

- ***Unknown Fraction of Members*** – In this case, the attacker does not know the proportion of training members in the candidate pool. The pool is constructed to contain $80k$ examples. Reference-model attacks that assume a 1:1 ratio train their reference models on $40k$ samples (half treated as members and half used to estimate the loss distribution of non-members), while the target model is trained on only $25k$. This mismatch causes the member/non-member loss distributions to differ significantly. We evaluate this scenario individually on CINIC-10, since the other datasets are too small for a meaningful setup.

- ***No-Auxiliary-Knowledge Setting*** – Combining the above cases, the candidate pool for each dataset has $80k$ samples formed by mixing $30k$ in-distribution with $50k$ OOD images. The attacker trains reference models on a $40k$-example subset sampled from the full mixed pool under different configurations and without access to the true member ratio.

# 6 Results

In this section, we quantitatively compare our *ImpMIA* attack against prominent prior black-box and white-box attacks across multiple datasets. In Section 6.1, we demonstrate the superiority of our attack in the realistic scenario where only the model weights are known and the attacker is given a candidate set of samples that includes the training data, but without any additional knowledge of the model training or the candidate set composition. Next, in Section 6.2, we systematically analyze the impact of eliminating each assumption across different models, showing that reference-model-based methods suffer significant performance drops, while our method remains unaffected.

## 6.1 Membership Inference Results

We report membership inference attack results across three datasets, demonstrating the superiority of our proposed attack in the realistic scenario described above (without any assumptions on model training, data distribution, or member ratio). As shown in Table 1, reference-model attacks such as LiRA and RMIA struggle severely in this regime: at 0.0% False Positive Rate (FPR), LiRA achieves only 0.17% True Positive Rate (TPR) on CIFAR-10, and RMIA 0.01%. Importantly, these are the best-performing attacks, even compared to white-box methods, when the knowledge assumptions are provided (see Table T10 in the appendix). Their effectiveness relies on training large sets of reference models under matching conditions, a requirement that breaks down when the attacker cannot replicate the target's training process. In contrast, *ImpMIA* avoids training reference models entirely, and maintains strong TPR at low FPR across all datasets. On CIFAR-10, our attack achieves 1.41% TPR at 0.0% FPR, and 2.76% at 0.01% FPR, with similar or larger gains on the other datasets. Our approach achieves substantially stronger performance at strict low-FPR operating points, which are the most relevant in practical membership inference scenarios, and comparable AUC results. ROC curves are shown for CIFAR-10 in Fig. 3 and for CIFAR-100 and CINIC-10 in Fig. S3. Importantly, our attack is computationally much cheaper than reference-based methods, as it does not require training reference models and is about 4× faster than reference-model attacks (see App. B.11). Finally, Table 2 shows that ImpMIA outperforms prior attacks on VGG16 and ResNet50 in the low-FPR regime.

## 6.2 Assumptions Influence Analysis

We present a detailed analysis of the influence of removing each assumption, individually and jointly, on attack performance. As discussed earlier, reference-model-based methods rely on training reference models under conditions that closely match the target model. Their effectiveness comes from approximating the loss distributions of members versus non-members, which requires alignment between target and reference training.

In Table 3 we report results on the CINIC-10 dataset across five scenarios: all assumptions provided (Full Auxiliary-Knowledge setting), the removal of each assumption individually, and the removal of all three assumptions simultaneously. As shown, all methods except our proposed *ImpMIA*, which does not rely on any reference models, suffer substantial drops in performance when all assumptions are removed. In particular, LiRA and Attack-R exhibit a clear degradation as assumptions are removed. While they perform well in the

| | VGG16 | | | ResNet50 | | |
|---|---|---|---|---|---|---|
| Attack | AUC | @0.01 % | @0.0 % | AUC | @0.01 % | @0.0 % |
| *LiRA* | 0.75 | 0.62 | 0.18 | 0.79 | 1.02 | 0.26 |
| *RMIA* | **0.78** | 0.69 | 0.06 | 0.78 | 0.09 | 0.02 |
| *Ours* | **0.78** | **1.59** | **0.56** | **0.82** | **1.80** | **0.78** |

Table 2: **Membership Inference results on different architectures (No-Auxiliary-Knowledge setting) .** Performance of LiRA, RMIA, and our method on VGG16 and ResNet50. We report AUC and TPR at fixed false-positive rates (FPR = 0.01% and 0.0%).

| Method | Full Auxiliary-Knowledge | Unknown Training Config. | Different Distribution | Unknown Fraction of Members | NO Auxiliary-Knowledge |
|---|---|---|---|---|---|
| *Attack-R* | 4.62 / 1.81 | 1.62 / 0.37 | 3.27 / 1.42 | 2.42 / 0.0 | 0.31 -93.3% / 0.0 -100% |
| *LiRA* | 7.59 / 5.03 | 5.81 / 3.47 | 3.73 / 1.84 | 5.70 / 2.82 | 2.27 -70.1% / 0.66 -86.9% |
| *RMIA* | 0.24 / 0.08 | 0.92 / 0.32 | 0.28 / 0.08 | 0.34 / 0.06 | 0.15 -37.5% / 0.03 -62.5% |
| **ImpMIA (ours)** | 3.67 / 2.28 | 3.67 / 2.28 | 3.28 / 2.69 | 5.19 / 1.96 | 5.32 / 2.47 |

Table 3: **Evaluation of assumptions influence.** CINIC-10 membership inference across five scenarios: Auxiliary-Knowledge Setting (all assumptions), removal of each assumption individually, and removal of all three simultaneously. Each entry shows TPR (%) at 0.01% / 0.0% FPR. The last column reports relative % of performance drop with respect to the Full Auxiliary-Knowledge Setting.

Full Auxiliary-Knowledge setting, their TPR at low FPR decrease substantially under unknown configuration, distribution shift, and unknown ratio. In contrast, our method achieves strong results in these settings, as it does not depend on such assumptions. We observed improvements in *ImpMIA* at 0.01% FPR level when the fraction of members was changed, which may be explained by the evaluation metric's dependence on the pool size (see App. B.5). It is worth noting that RMIA's performance in the Full Auxiliary-Knowledge setting is lower than those reported in its original paper. This is because our training follows the white-box setup from Cohen & Giryes (2024), which differs from RMIA's original experimental setting. This may imply limited generalization of RMIA (see App. C.1 for further discussion).

### 6.3 Ablation and Analysis

The Appendix presents additional ablation studies and analyses. This includes: visualizations of the $\lambda$ values (App. B.1); applicability to large candidate pools (App. B.2); performance under partial training-set coverage (App. B.3); the effect of pool size on evaluation (App. B.5); ablations on the scoring and aggregation components (App. B.6), the effect of weight decay (App. B.9) and pre-filtering strategies (App. B.10), and running time analysis (App. B.11).

## 7 Conclusion

*ImpMIA* advances Membership Inference by introducing a simple theory-driven white-box attack which leverages the implicit bias in neural-nets. It outperforms both black- and white-box baselines in scenarios where the attacked network's training procedure & settings are unknown. By resorting to pure optimization of KKT conditions on the attacked model's parameters (without the need to train any reference models), our method allows efficient scaling to very large candidate pools (see App. B.2). The white-box assumption is increasingly realistic as many modern models are publicly released with their full parameters. *ImpMIA* is best suited for scenarios where the training-set is contained in the examined-set. However, we show that this assumption can be relaxed (App. B.3), and that *ImpMIA* is competitive even at low coverage of the training-set (only 10%), and becomes superior at high coverage. Our work is the first to demonstrate the implications of implicit bias theory to membership inference attacks. More broadly, it provides a concrete case study of how insights from implicit bias theory, which have largely been developed in idealized or small-scale settings, can be instantiated in practical machine learning tasks, going beyond theory and toy examples to real datasets, larger neural networks, and standard training regimes. *ImpMIA* provides a step forward in practical privacy auditing, establishing a bridge between implicit bias theory and applied machine learning.

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

# Appendix

## A  ImpMIA Additional Implementation Details

Algorithm 1 provides a self-contained summary of the complete ImpMIA attack pipeline.

---

**Algorithm 1** ImpMIA Attack Pipeline

---

**Input:** Trained parameters $\theta$; candidate set $\mathcal{X} = \{(x_i, y_i)\}_{i=1}^{M}$; augmentations $\mathcal{T}$; number of parameter blocks $K$
**Output:** Membership scores $\{s_i\}_{i=1}^{M}$
1: Apply augmentations in $\mathcal{T}$ to the candidate set $\mathcal{X}$
2: Remove misclassified samples from $\mathcal{X}$
3: **for all** remaining samples $x_i$ and augmentations $t \in \mathcal{T}$ **do**
4:     Compute the margin gradient
       $g_i^{(t)} \leftarrow \nabla_\theta m_i^{(t)}(\theta)$
5: **end for**
6: Partition the model parameters into $K$ layer/filter-wise blocks $B_1, \ldots, B_K$
7: **for** $k = 1, \ldots, K$ **do**
8:     Restrict the parameters and gradients to block $B_k$
       $\theta_k \leftarrow \theta|_{B_k}, \quad g_{i,k}^{(t)} \leftarrow g_i^{(t)}|_{B_k}$
9:     Construct the block gradient matrix $A_k$ from $\{g_{i,k}^{(t)}\}_{i,t}$
10:     Center and normalize $A_k$ and $\theta_k$
11:     Solve for the block-wise coefficients
       $\lambda_k^\star \leftarrow \arg\min_\lambda \left[ 1 - \cos(A_k\lambda, \theta_k) + \alpha L_{\text{neg}}(\lambda) + \beta L_{\text{marg}}(\lambda) \right]$
       where $L_{\text{neg}}$ penalizes $\lambda_i < 0$, and $L_{\text{marg}}$ is an inverse-margin penalty on $|\lambda_i|$.
12:     Store $\lambda_k^\star$
13: **end for**
14: Aggregate recovered $\lambda$ values across parameter blocks and augmentations
15: Apply margin-based post-processing
16: **return** final membership scores $\{s_i\}_{i=1}^{M}$

---

The overall pipeline includes the following parts: (i) **pre-filtering** of candidate samples based on their classification margin, (ii) **augmentation** using horizontal flips, (iii) **block division** of weights for more efficient optimization, (iv) **gradient matrix construction** per block-structured gradient matrix $A$ over the chosen model parameters, (v) **optimization** to solve the KKT conditions and learn the coefficients $\lambda$ via blockwise optimization with dedicated regularization, (vi) **coefficient aggregation** across blocks into robust per-sample scores, and finally (vii) **post-processing** of the coefficient scores to derive the final per-sample score.

We first perform **pre-filtering**, where we filter out misclassified samples, since training members are more likely to be correctly predicted. For each candidate sample $(x_i, y_i)$, we compute its logit margin:

$$\Delta_i = \Phi_{y_i}(\theta; x_i) - \max_{j \neq y_i} \Phi_j(\theta; x_i),$$

and discard those with $\Delta_i < 0$. For each remaining sample we apply **augmentation**, including both the original and its horizontal flip. This reflects the fact that most models are trained with augmentations, while keeping computational cost low by not adding further augmentations. Final scores are averaged across the augmented views of each sample.

Next is **block division**. To manage dimensionality and improve optimization, parameters are partitioned into blocks of $\sim 1.5 \times 10^5$ entries and solved one block at a time. For CIFAR-10 and CINIC-10 we include all layers, while for CIFAR-100 we restrict to the final convolutional stages, where membership signals are strongest (Nasr et al., 2019). Parameters are grouped by layer order, and filters inside each convolutional layer are grouped together, since weights from the same filter/layer share statistical properties such as sparsity and magnitude. This improves conditioning of the system.

In the **gradient matrix construction** step, for each retained sample we compute the gradient of its margin w.r.t. the parameters and stack them as columns of a matrix $A \in \mathbb{R}^{p \times M}$. Both the gradient block and the target parameter vector are centered and normalized.

In the **optimization** step, for each block $b$ with parameters $\theta_{(b)}$, we solve for coefficients $\lambda_{(b)}$ by minimizing:

$$\mathcal{L} \;=\; 1 - \cos\big(A_{(b)}\lambda_{(b)}, \theta_{(b)}\big) + \alpha\,\mathcal{L}_{\mathrm{neg}} + \beta\,\mathcal{L}_{\mathrm{marg}}.$$

The constants $\alpha, \beta \geq 0$ are regularization weights controlling the strength of $L_{\mathrm{neg}}$ and $L_{\mathrm{marg}}$, respectively. $L_{\mathrm{neg}}$ penalizes negative entries in $\lambda^{(b)}$, encouraging the non-negativity constraint of the KKT multipliers. The margin regularizer is implemented as an inverse-margin weighted $\ell_1$ penalty, $L_{\mathrm{marg}} = \frac{1}{M}\sum_i |\lambda_i^{(b)}|/d_i$, where $d_i = \max(\Delta_i, \epsilon)$ is the clipped distance of sample $i$ from the decision boundary. Thus, this term penalizes large coefficients assigned to very low-margin samples, stabilizing the optimization and reducing spurious large coefficients near the decision boundary. We use cosine similarity because it removes scale sensitivity and is more robust than $\ell_2$ loss, preventing non-members from being incorrectly emphasized. Optimization uses AdamW with cosine learning-rate scheduling, gradient clipping, and early stopping. Coefficients are debiased by stored column norms and $z$-scored within each block.

After optimizing each block separately, we perform **coefficient aggregation**. Each block $j$ yields a coefficient vector $\lambda^{(j)}$. For each sample $i$, we collect $\{\lambda_i^{(j)}\}_j$, sort them, and fuse into a robust score using a trimmed mean (averaging central values while discarding extremes) and a signal-to-noise ratio (SNR, mean over standard deviation across blocks). This suppresses spurious outliers and emphasizes consistent member signals.

Finally, in the *post-processing* step, we refine the raw scores using three components: **(i) class-level margin boosting, (ii) sample-level margin boosting, and (iii) distance-based scaling**. Both boosting mechanisms adjust a sample's score according to its margin (distance from the decision boundary), but at different granularities (class vs. individual sample). **(i) Class-level boosting** is a simple per-class rescaling step, motivated by prior work (Nasr et al., 2019) showing that some classes are systematically more vulnerable to membership inference. Our boost builds on the observation that harder classes (with lower average margins) are more likely to memorize additional samples, so an attack can potentially recover more members from these classes. Accordingly, we assign slightly higher weight to harder classes and mildly downweight easier ones via a small per-class scaling coefficient of 0.03. **(ii) Sample-level margin boosting** then refines scores within each class: following previous observations (e.g., Haim et al.) that near-boundary points are more likely to be memorized, we slightly upweight low-margin, near-boundary samples and mildly downweight very high-margin ones, using a per-sample scaling factor of 0.3. **(iii) Distance-based scaling** further penalizes deviation from the estimated class margin. We approximate the class margin $\bar{m}_c$ as the mean margin of the top-$k$ highest-coefficient samples in each class, and rescale scores by dividing by $|\Delta_i - \bar{m}_c|^\eta$, where $\Delta_i$ is the margin of sample $i$ and $\eta > 0$ controls how strongly this distance affects the rescaling (we set $\eta = 0.05$ in all experiments). This ensures that only samples whose margins are consistent with typical member behavior retain high scores, helping to reduce false positives among non-members.

These practical choices address two main difficulties. First, the full gradient system is extremely large, since the matrix $A$ has one row per selected parameter and one column per retained candidate sample. Solving the full system directly is therefore memory-intensive and poorly scalable. We address this by optimizing the coefficients block-wise, which reduces peak memory and allows the method to scale to larger networks and candidate pools. Second, different layers and filters have different statistical properties, such as scale, sparsity, and gradient magnitude. Mixing all parameters in a single system can therefore make convergence harder. We address this by grouping parameters according to layer/filter structure, centering the gradient block and target parameter vector, normalizing gradient columns, and later debiasing the recovered coefficients using the stored column norms. We also optimize a cosine-similarity reconstruction objective rather than a standard $\ell_2$ objective, since cosine similarity reduces sensitivity to scale differences across blocks and improves optimization stability. These steps are not part of the formal KKT equations, but are numerical stabilization choices that make the KKT-inspired optimization feasible in practice. This is especially important when moving beyond the shallow MLP settings studied in prior implicit-bias reconstruction works to larger convolutional networks and large candidate pools.

**Layer Selection for CIFAR-100.** For CIFAR-10 and CINIC-10 we construct the gradient matrix $A$ from all convolutional layers of the network. For CIFAR-100, however, we restrict $A$ to the final convolutional stages. This choice is motivated by both prior work and empirical observations. Nasr et al. (2019) empirically observed that, in deep CNNs trained on CIFAR-100, the strongest membership signals reside in the gradients of the later layers, while earlier layers mostly learn generic low- and mid-level features that generalize well and leak relatively little membership information. In a many-class regime such as CIFAR-100, early and mid-level layers must be heavily shared across classes, whereas the final convolutional blocks and classifier head implement sharp, class-specific decision boundaries and absorb most of the memorization.

**Hyperparameter Selection.** All hyperparameters of *ImpMIA* were selected using a held-out evaluation set. For CIFAR-10 and CINIC-10 we used a dedicated validation split constructed from CINIC-10, which is large enough to provide an independent validation pool. For CIFAR-100 we used a validation split taken from the generated candidate set. For each dataset, we followed the same protocol as in our main experiments: we trained 5 target models with different random seeds (affecting both the data split and the initialization) under the realistic No-Auxiliary-Knowledge setting, and ran *ImpMIA* for a small grid of hyperparameter configurations. We then selected the configuration that achieved the best average performance across these 5 runs according to our main evaluation criteria (TPR at low FPR). The same hyperparameters are used across all scenarios and across all architectures (ResNet-18, VGG16, and ResNet50).

# B  Additional Ablation and Analysis

In this section, we provide further ablations and analyses of our approach: (i) visualization of the $\lambda$ scores across different classes (App. B.1); (ii) scalability to large candidate pools (App. B.2); (iii) the effect of training-set coverage in the evaluated set (App. B.3); (iv) accuracy of the unknown training configuration models (App. B.4); (v) the effect of the non-member ratio and pool size on evaluation (App. B.5); (vi) ablations of the scoring and aggregation variants (App. B.6); (vii) the effect of block construction (App. B.7); (viii) the effect of training duration (App. B.8); (ix) the influence of weight decay (App. B.9); (x) the influence of pre-filtering on performance (App. B.10); and (xi) an analysis of the running time (App. B.11).

## B.1  Visualization of $\lambda$ Scores.

In Fig. S1, we present results for six different CIFAR-100 classes. The plots show the $\lambda$ score on the y-axis and the distance from the decision boundary on the x-axis. As expected, high $\lambda$ scores are strong indicators of membership. Samples very close to the decision boundary are more likely to be non-members, while almost all members with high $\lambda$ values fall slightly farther from the decision boundary. This is consistent with the fact that models are trained to push training samples away from the decision boundary creating a margin, while hard test samples can lie closer to it. Interestingly, training samples that remain close to the margin are those that the network tends to memorize (Haim et al., 2022), and therefore receive higher $\lambda$ scores, reflecting their strong influence on the model's predictions and weights.

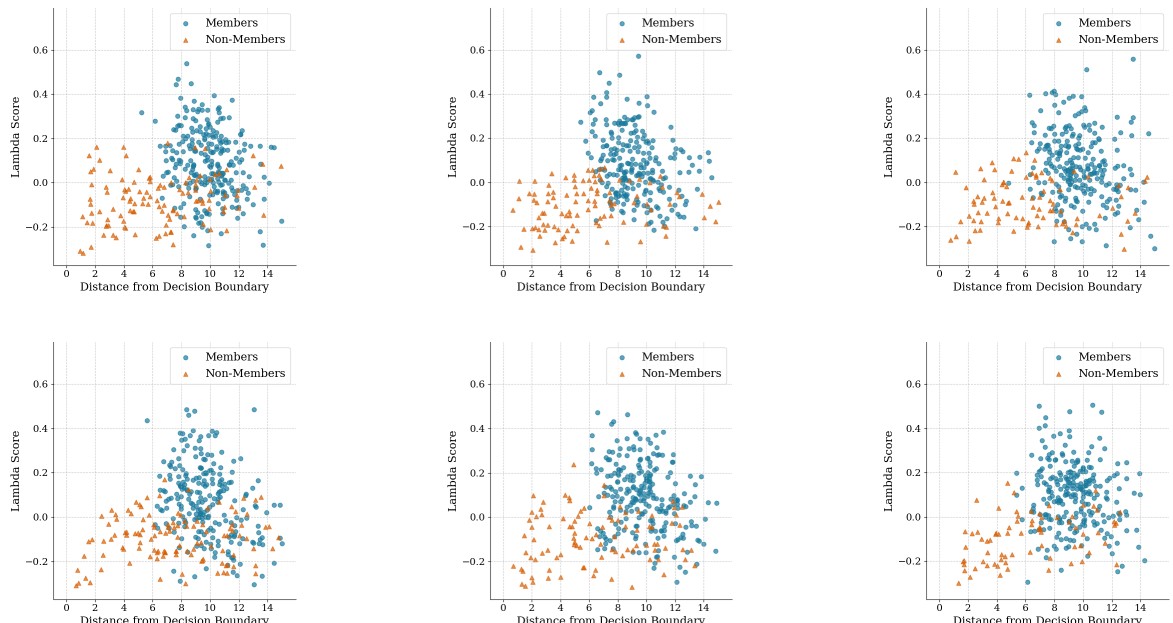

Figure S1: **Lambda scores visualization (CIFAR-100).** Scatter plots for six different classes. Each plot shows the evaluation samples; x-axis is distance from the decision boundary, y-axis is the $\lambda$ score, and points are colored by membership (member vs. non-member). High $\lambda$ values strongly indicate membership.

## B.2 Large Candidate Pool

We further evaluated ImpMIA under larger candidate pools. In addition to the 50,000 ($\approx 2\times$) and 80,000 ($\approx 3\times$) candidate sets reported in the main text, we consider a pool of 250,000 candidates ($\approx 10\times$ the training size) on CIFAR-10. Following our realistic no auxiliary knowledge protocol, the target model is trained on 25,000 CIFAR-10 samples, while the reference set consists of the remaining 25,000 CIFAR-10 samples together with 200,000 images from CINIC-10. As shown in Table T1, ImpMIA achieves TPRs of 3.00% at 0.01% FPR and 0.54% at 0.0% FPR, substantially outperforming other MIAs at low FPRs in the larger candidate pool scenario.

| Attack | AUC | @0.01 % | @0.0 % |
|--------|-----|---------|--------|
| *Attack-R* | 0.75 | 0.01 | 0.0 |
| *Attack-P* | 0.76 | 0.12 | 0.0 |
| *LiRA* | **0.80** | 0.54 | 0.11 |
| ***Ours*** | 0.79 | **3.00** | **0.54** |

Table T1: **Membership inference with a 250K candidate pool (CIFAR-10).** We compare ImpMIA to other MIA baselines under the realistic No-Auxiliary-Knowledge setting. We report AUC and TPR at fixed false-positive rates (FPR = 0.01% and 0.0%).

## B.3 Effect of Training Sample Coverage in the Evaluation Set

Our method relies on the implicit bias of the network, linking training samples to the learned weights. Consequently, if a large portion of the training set is missing from the evaluation set, performance naturally decreases. Importantly, in practice it is reasonable to expect evaluation set that cover most of the training set, especially since our method's efficiency allows scaling to very large candidate pools without the need to train reference models. As shown in Table T2, for the No-Auxiliary-Knowledge setting performance is strongest when most of the training set is included, but our attack also works well under partial training coverage. In all cases, evaluation is on a random $8,000$ (10%) samples from the candidate pool to avoid influence from candidate pool size on the reported metrics.

In Table T3 we show a direct comparison with the previous best methods under only 10% coverage in our realistic scenario. Even with this low coverage, our results are comparable to the best competitors, and with higher coverage our method clearly surpasses them (Table 1). Higher coverage is a likely scenario in many practical settings, such as auditing whether a specific dataset was used to train a model, or when a dataset is known to contain the training set but the exact subset used is unknown..

| | CIFAR-10 | | |
|---|---|---|---|
| **Coverage** | **AUC** | **@0.1 %** | **@0.0 %** |
| *100%* | 0.81 | 6.80 | 2.26 |
| *75%* | 0.81 | 5.49 | 1.42 |
| *50%* | 0.81 | 5.34 | 1.61 |
| *25%* | 0.79 | 3.49 | 0.96 |
| *10%* | 0.75 | 3.01 | 0.94 |

Table T2: **Effect of training sample coverage (no auxiliary knowledge).** Ablation study on CIFAR-10 showing the impact of training sample coverage within the candidate superset. Performance is strongest when most of the training set is included, but our method remains effective under partial coverage.

| Attack | AUC | @0.1 % | @0.0 % |
|--------|-----|--------|--------|
| *Attack-P* | 0.76 | 0.38 | 0.06 |
| *Attack-R* | 0.74 | 1.73 | 0.32 |
| *LiRA* | **0.80** | 2.98 | **1.20** |
| *RMIA* | **0.80** | 2.43 | 0.14 |
| ***Ours*** | 0.75 | **3.01** | 0.94 |

Table T3: **Comparison under limited coverage of 10% (no auxiliary knowledge).** We compare our method on CIFAR-10 with previous attacks when the candidate set covers only 10% of the training data in our realistic No-Auxiliary-Knowledge setting. Even under this low coverage, our results are comparable to the best competitors.

### B.4 Accuracy of the Unknown Training Configuration Models

We verify that the models trained with the Unknown Training Configuration are not undertrained. As shown below, this configuration achieves nearly perfect train accuracy and strong test accuracy, close to the standard configuration. Both configurations converge well, indicating that the degradation of reference-model attacks is not caused by weak or undertrained reference models.

| | CIFAR-10 | |
|---|---|---|
| **Training configuration** | **Train accuracy** | **Test accuracy** |
| *Standard configuration* | 100.0% | 89.9% |
| *Unknown Training Configuration* | 99.9% | 88.0% |

Table T4: **Accuracy of the Unknown Training Configuration Models.** We compare the standard configuration with the configuration used in the Unknown Training Configuration setting.

### B.5 Influence of Non-Member Ratio on Evaluation

Our assumption-removal analysis also showed the influence of candidate pool size on model performance. In this setting, we lowered the member ratio by adding more non-members to the candidate set (increasing from 25K to 55K non-members). While our method does not rely on any specific member-to-non-member ratio, adding more non-members can affect results in two ways: (i) it increases the number of samples, which may make it harder to classify members, and (ii) it directly affects the evaluation metric, since FPR is defined relative to the total number of non-member samples. The overall performance reflects the interaction of these two factors. On the one hand, a larger pool introduces more distractors, potentially reducing accuracy. On the other hand, at fixed FPR thresholds (e.g., 0.01%), a larger pool allows more absolute mistakes. For instance, with 25K non-members, 0.01% FPR corresponds to only two false positives, whereas with 55K non-members, it allows up to five. Since coefficients are not uniformly distributed, permitting more mistakes can yield a nonlinear gain in TPR. In practice, we observed that at 0.01% FPR, our method performed better with 55K non-members than with 25K. This suggests that the positive effect of the evaluation metric outweighs the negative impact of additional distractors. Importantly, at 0.0% FPR, where the metric is unaffected by pool size, the results remained stable, as expected.

### B.6 Scoring Variants Ablation

We present an ablation study of the different score-refinement components in our pipeline for CIFAR-10 in our No-Auxiliary-Knowledge setting (see Table T5). After block-wise optimization, each sample has multiple coefficient estimates, one per block, which are fused into a robust score using both a trimmed mean (to discard extreme outliers) and a signal-to-noise ratio (SNR, mean over standard deviation across blocks). This aggregation step already improves robustness by emphasizing consistent membership signals. On top of this, we evaluate margin-based boosting and distance scaling. Class-level boosting gives higher weight to harder classes (with lower average margins), while per-sample boosting highlights points near the decision boundary, which are more likely to be memorized. Finally, distance scaling penalizes samples whose margins deviate from the estimated class margin, reducing false positives. As shown, each component contributes to performance gains, and the full *ImpMIA* pipeline achieves the strongest low-FPR detection. Moreover, with either aggregation option, and even without any post-processing, our attack already outperforms previous attacks and shows strong results. The post-processing steps have only a comparatively small influence at low FPR (e.g., TPR@0.0% ranges only from 1.23 to 1.41), while aggregation has a larger effect. This is expected, since aggregation determines how we exploit the main signal produced by the KKT-based optimization. Overall, the primary source of our gains comes from the implicit-bias-based optimization itself rather than from the specific post-processing.

| Variant | AUC | @0.01 % | @0.0 % |
|---|---|---|---|
| *Trimmed mean only* | 0.74 | 2.58 | 1.18 |
| *Robust SNR only* | **0.85** | 2.51 | 0.89 |
| *Fusion (trimmed mean + SNR)* | 0.82 | 2.60 | 1.23 |
| *+ Boost1 (class-level margins)* | 0.82 | 2.59 | 1.23 |
| *+ Boost2 (sample-level margins)* | 0.81 | 2.69 | 1.39 |
| *+ Trim division* | 0.82 | **2.84** | 1.31 |
| ***ImpMIA*** | 0.81 | 2.76 | **1.41** |

Table T5: **Ablation of ImpMIA components for CIFAR-10 (no auxiliary knowledge).** Performance of different scoring variants and incremental boosts. We report AUC and TPR at fixed false-positive rates (FPR = 0.01% and 0.0%).

## B.7 Block Construction

Different layers can have different parameter and gradient statistics, which can make the coefficient optimization less stable when unrelated parameters are mixed in the same block. This motivates our default sequential block construction, where nearby parameters, usually from the same or nearby layers, are grouped together. We observe this effect empirically on a subset of the candidate set using smaller parameter batches. Sequential blocks have more homogeneous statistics: the row-norm coefficient of variation is (0.48) for sequential blocks versus (0.91) for random blocks, and the mean absolute row correlation is (0.163) versus (0.069). This suggests that sequential blocks group parameters with more similar gradient statistics. We compare our default sequential blocks with a random-block variant, where parameters are randomly assigned to blocks while keeping all other components unchanged. As shown in Table T6, sequential blocks perform much better in the low-FPR regime. This supports our choice to group parameters with similar gradient statistics, leading to more stable and effective coefficient recovery.

| | CIFAR-10 | | |
|---|---|---|---|
| Block construction | AUC | @0.01 % | @0.0 % |
| *Sequential blocks* | **0.81** | **2.76** | **1.41** |
| *Random blocks* | 0.80 | 0.37 | 0.08 |

Table T6: **Effect of Block Construction on ImpMIA.** We compare random block construction with our default sequential block construction. Sequential blocks perform better, especially in the low-FPR regime.

## B.8 Effect of Training Duration

We evaluate how ImpMIA changes with the number of training epochs. This directly tests whether the attack improves as training moves closer to the long-time regime assumed by the implicit-bias motivation. As shown in Table T7, ImpMIA improves with longer training: AUC increases from (0.65) at 20 epochs to (0.81) at 400 epochs, while TPR at (0.0%) FPR increases from (0.04%) to (1.41%). This supports the connection between our KKT-inspired signal and the long-time implicit-bias regime.

| | CIFAR-10 | | |
|---|---|---|---|
| Training epochs | AUC | @0.01 % | @0.0 % |
| *20* | 0.65 | 0.08 | 0.04 |
| *100* | 0.78 | 1.46 | 0.87 |
| *400* | 0.81 | 2.76 | 1.41 |
| *1000* | 0.81 | 3.01 | 1.55 |

Table T7: **Effect of Training Duration.** Ablation on CIFAR-10 showing the effect of the number of training epochs on ImpMIA performance. Longer training improves the attack, especially in the low-FPR regime.

### B.9 Weight Decay Influence

To study the effect of explicit weight decay, we evaluated *ImpMIA* on CIFAR-10 No-Auxiliary-Knowledge setting, both with the standard decay level ($10^{-4}$) and without decay. While the KKT stationarity formulation applies in the homogeneous case, prior theory suggests that in non-homogeneous networks explicit weight decay is needed as a replacement. However, our results show that even in this setting the attack remains effective without weight decay: the no-weight decay variant achieves the same AUC (0.81) and slightly stronger performance in the low-FPR regime (2.93 vs. 2.76 at 0.01% FPR; 1.62 vs. 1.41 at 0.0% FPR). This is consistent with the intuition that stronger regularization can suppress memorization and weaken membership signals. Overall, this demonstrates the robustness of our method and that weight decay is not strictly necessary for strong empirical performance.

| | CIFAR-10 | | |
| --- | --- | --- | --- |
| **Variant** | **AUC** | **@0.01 %** | **@0.0 %** |
| *Without weight decay* | 0.81 | 2.93 | 1.62 |
| **$10^{-4}$** | 0.81 | 2.76 | 1.41 |

Table T8: **Effect of Weight Decay (no auxiliary knowledge).** Ablation on CIFAR-10 comparing the standard weight decay level ($10^{-4}$) to training without weight decay. Although implicit bias theory assumes weight decay in the non-homogeneous case for the KKT characterization, our results show the attack remains effective regardless.

### B.10 Influence of Pre-Filtering

The pre-filtering step aims to both stabilize the optimization by eliminating samples that are unlikely to be members (and therefore may add noise) and speed up optimization by having fewer samples. On the other hand, it may discard data points that are hard to learn even though they were in the training data, and therefore miss member samples. Since any attack is likely to miss some members, this is a trade-off that can be chosen in practice.

We analyze the impact of the pre-filtering step on our method's performance in the CIFAR-10 No-Auxiliary-Knowledge setting. On CIFAR-10, pre-filtering removes between 4–25% of the examples, depending on the seed and scenario. As shown in Table T9, using pre-filtering leads to a slight increase in AUC (from 0.78 to 0.81), as well as improvements in TPR at 0.01% FPR (from 2.28 to 2.76) and at 0.0% FPR (from 1.15 to 1.41). Overall, the pre-filtering step does not appear to have a significant influence on performance, but it does speed up optimization by reducing the number of points being optimized

| Variant | AUC | @ 0.01% FPR | @ 0.0% FPR |
| --- | --- | --- | --- |
| No pre-filtering | 0.78 | 2.28 | 1.15 |
| With pre-filtering | 0.81 | 2.76 | 1.41 |

Table T9: **Effect of pre-filtering on ImpMIA (CIFAR-10, no auxiliary knowledge).** Comparison of AUC and TPR at very low FPR with and without the pre-filtering step. We report AUC and TPR at fixed false-positive rates (FPR = 0.01% and 0.0%).

### B.11 Runtime and Complexity Analysis

**Complexity and memory.** We first analyze the computational cost of ImpMIA. Several parts of the pipeline are relatively negligible compared to the main computation. Margin computation and pre-filtering require only forward passes, while score aggregation and post-processing are simple operations over the recovered coefficients. The two dominant costs are: (i) computing per-sample margin gradients, and (ii) the block-wise coefficient optimization. Let $M$ denote the number of retained candidate samples after filtering, $|\mathcal{T}|$ the number of augmentations, $p$ the number of selected parameters, $K$ the number of parameter blocks, $b \approx p/K$ the number of parameters per block, and $I$ the number of optimization iterations per block. Gradient construction requires one backward pass per candidate sample and augmentation, and therefore

scales linearly with $M$ and $|\mathcal{T}|$. For the optimization, each block uses a gradient matrix of size $b \times M|\mathcal{T}|$, so each optimization iteration scales linearly with $b$, $M$, and $|\mathcal{T}|$. Since we optimize over $K$ blocks, the total optimization cost scales linearly with $I$, $p$, $M$, and $|\mathcal{T}|$. The peak memory is controlled by the block size, since blocks are processed sequentially. Thus, memory scales with $bM|\mathcal{T}|$ rather than $pM|\mathcal{T}|$. In practice, this block-wise formulation allows us to process about 100K candidates on a single H100 GPU with 80GB memory. Scalability can be further improved by filtering samples far from the decision boundary, since such samples are unlikely to influence the KKT equations or receive large coefficients.

**Empirical runtime.** The runtime of *ImpMIA* ranges between 12–16 hours on a single H200 GPU, depending on the size of the candidate pool (50K–80K samples), and can be reduced to approximately 2 hours using 8 GPUs. The black-box reference-model attacks in their strongest setting, used in this paper as well, require training 256 reference models. In the Full Auxiliary-Knowledge setting (400 epochs with a candidate pool of size 50K), this results in a total runtime of approximately 48 hours on a single GPU, which is roughly 4× slower than *ImpMIA*. Notably, the runtime of black-box reference-model attacks is highly influenced by the number of epochs assumed by the attacker to mimic the target model's training (which can be both lower or much higher than 400). In contrast, the runtime of *ImpMIA* remains largely unchanged, as it requires no reference model training.

## C  Competitive Baselines: Technical Details

### C.1  Black-Box Baselines

For black-box comparisons, we used the official **RMIA** implementation, which also includes code for **LiRA**, **Attack-P**, and **Attack-R**. We ran the *full-power* RMIA variant described by Zarifzadeh et al. (2023), which trains 256 reference models per dataset, and used the same framework to evaluate the other black-box baselines. To ensure consistency, we adapted the code to match the training configuration of Cohen & Giryes (2024) (SIF): ResNet-18 backbone, inputs normalized to $[0, 1]$, and standard augmentations (random crop and horizontal flip).

RMIA requires dataset-specific configurations, which we followed exactly as provided in their code and paper for CIFAR-10, CIFAR-100, and CINIC-10. All experiments were therefore run with the recommended hyperparameters for each dataset. We emphasize that RMIA, despite reporting state-of-the-art results in its original paper, is highly sensitive to training configurations, especially at very low false-positive rates. Even minor mismatches in normalization, architecture, optimizer, or learning-rate schedule caused severe degradation in our experiments.

LiRA distinguishes between two reference-model settings, commonly referred to as "online" and "offline". In the online setting, the candidate sample whose membership is being evaluated is known when the reference models are constructed. The attack can therefore compare the target-model behavior to two reference distributions: one estimated from reference models trained with that candidate sample, and one estimated from reference models trained without it. In the offline setting, the reference models are trained before the evaluated candidate set is given, using a separate set of samples. The attack therefore estimates only the non-member behavior from these independently trained reference models.

Both online and offline variants can be applied in our set-based evaluation, including the No-Auxiliary-Knowledge setting. However, both remain reference-model-based attacks: they require auxiliary reference models intended to mimic the target model's behavior, and therefore still depend on target-like training hyperparameters, data distribution, and candidate-set composition. Thus, while the offline variant reduces the amount of per-candidate information used by the attack, it does not remove the auxiliary-knowledge assumptions studied in our paper.

We report the online variants in the main paper because they are the stronger and more favorable baselines. For completeness, we also report the offline variants in Table T14. As shown there, the offline version of LiRA performs worse in the No-Auxiliary-Knowledge setting.

### C.2  White-Box Baselines

**Gradient-based Baselines.**  (Nasr et al., 2019) showed that the most informative white-box membership signal is the magnitude of per-sample gradients, since stochastic gradient descent drives member gradients toward zero. Building on this, we evaluate two baselines: (i) a *loss-gradient* score based on the gradients of the loss $\|\nabla_\theta L(f(x), y)\|$, and (ii) a *margin-gradient* score based on the gradients of the margin $\|\nabla_\theta[f_y(x) - \max_{j \neq y} f_j(x)]\|$. For each sample (and its horizontal flip), we compute per-layer gradient norms, convert them into rank-based scores (higher rank = smaller norm), and average across layers and augmentations. The loss-gradient baseline directly instantiates Nasr et al. (2019) observation, while the margin-gradient baseline adapts it to sensitivity around the margin, making it conceptually closer to *ImpMIA*.

**SIF Attack.**  The self-influence function (SIF) attack of Cohen & Giryes (2024) achieved state-of-the-art white-box membership inference in their paper, particularly under strong augmentations where gradient-norm methods fail. SIF measures a sample's effect on its own loss by approximating the influence function via recursive Hessian–vector products. Membership is inferred by whether a correctly classified sample's score lies close to zero, since training points tend to cluster tightly around this value while non-members yield more extreme values.

The original SIF formulation assumes access to labeled calibration samples to set the decision thresholds, and results were reported only on small subsets of data (500 calibration, 2,500 evaluation), producing binary

member/non-member predictions. In our setting, however, no labeled calibration samples are available, and we require continuous membership scores to compute TPR at low FPR across the full evaluation pool. We therefore adapt the method with a simple *distance-to-zero* criterion: the closer a score is to zero, the stronger the evidence of membership. To stabilize the Hessian inversion, we use four recursion steps and average over four stochastic estimates. This reduced setting makes the attack faster while remaining stable across the full evaluation pool.

### C.3 Additional Results

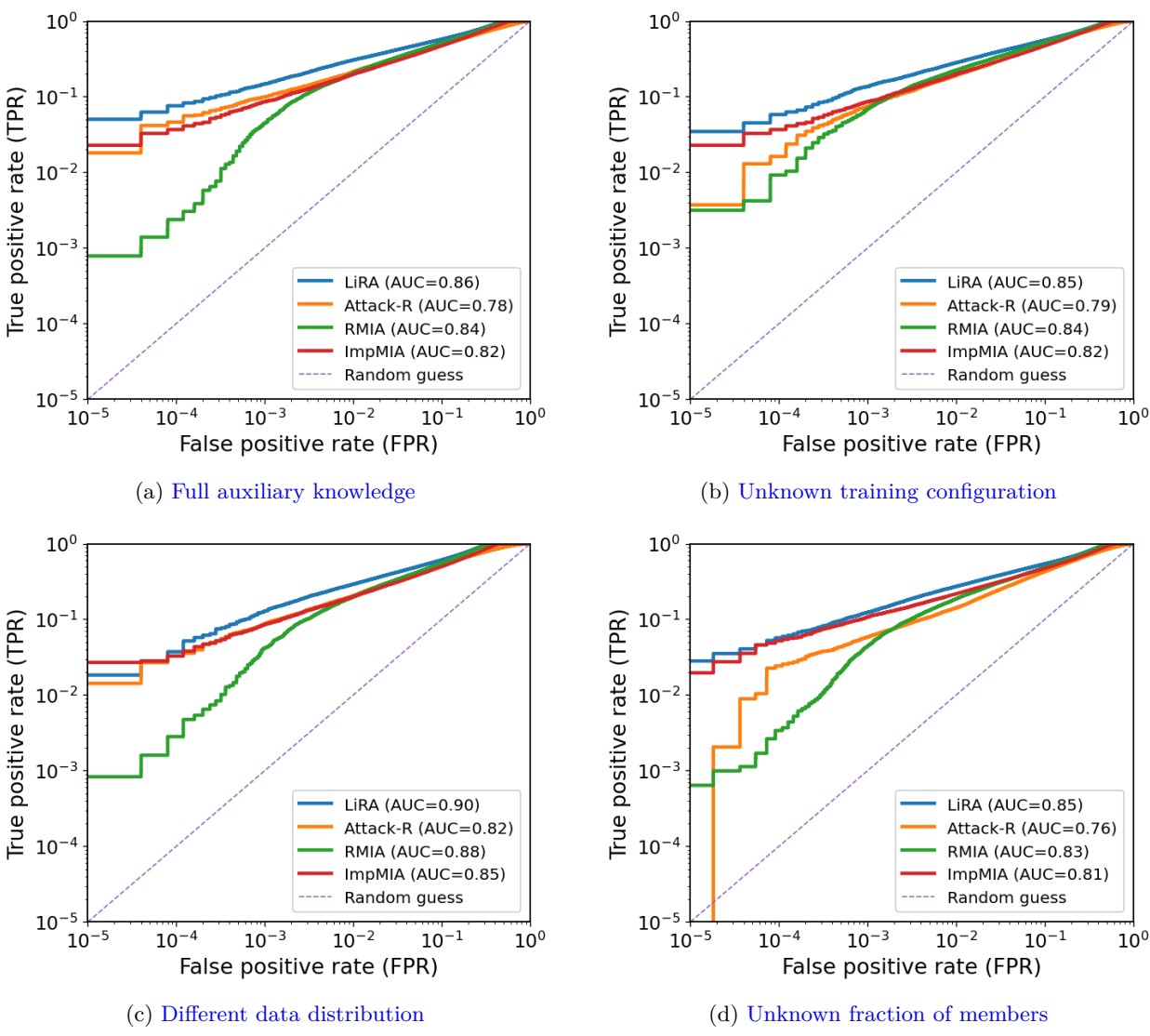

(a) Full auxiliary knowledge

(b) Unknown training configuration

(c) Different data distribution

(d) Unknown fraction of members

Figure S2: **TPR–FPR curves for CINIC-10 under different auxiliary-knowledge settings.** We report ROC curves for the compared attacks in the Full Auxiliary-Knowledge setting and in settings where one auxiliary assumption is removed: unknown training configuration, different data distribution, and unknown fraction of members. The plots complement the tabular results by showing the full ROC behavior, especially in the low-FPR regime.

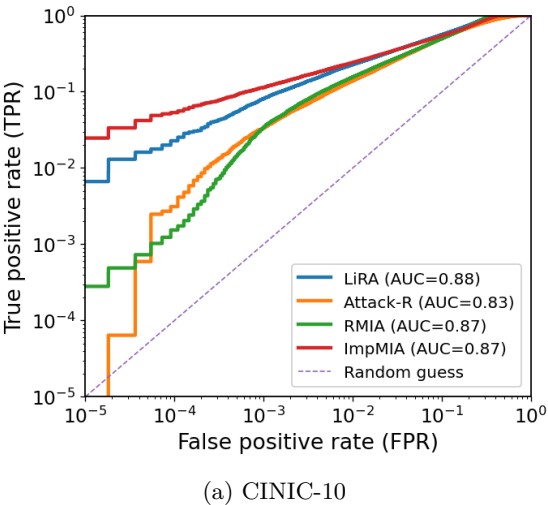

(a) CINIC-10

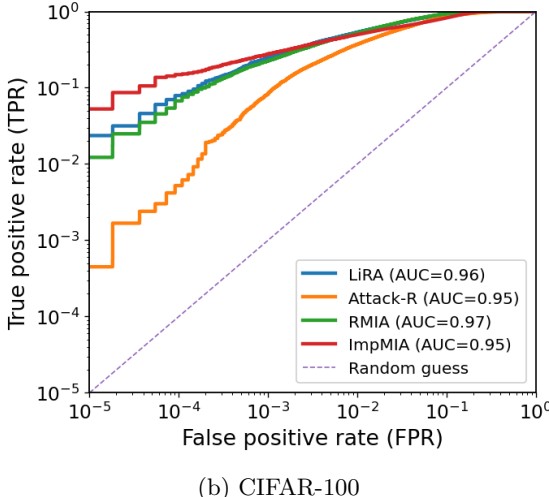

(b) CIFAR-100

Figure S3: **TPR–FPR plots for the No-Auxiliary-Knowledge setting.** These curves illustrate attack performance when the attacker faces realistic uncertainty: (i) training hyperparameters are unknown, (ii) the candidate pool mixes in- and out-of-distribution samples (distribution shift), and (iii) the fraction of members is unknown. The plots complement the main text by showing the full ROC behavior, especially in the low-FPR regime.

| Attack | CIFAR-10 | | | CIFAR-100 | | | CINIC-10 | | |
|---|---|---|---|---|---|---|---|---|---|
| | AUC | @0.01 % | @0.0 % | AUC | @0.01 % | @0.0 % | AUC | @0.01 % | @0.0 % |
| *Attack-P* | 0.59 | 0.01 | 0.0 | 0.81 | 0.0 | 0.0 | 0.70 | 0.0 | 0.0 |
| *Attack-R* | 0.67 | 2.56 | 1.58 | 0.88 | 19.04 | 16.07 | 0.78 | 4.62 | 1.81 |
| *LiRA (online)* | **0.75** | **5.28** | **3.56** | **0.95** | 24.26 | 15.25 | **0.86** | **7.59** | **5.03** |
| *LiRA (offline)* | 0.58 | 1.94 | 0.74 | 0.81 | 5.88 | 2.44 | 0.69 | 2.59 | 1.17 |
| *RMIA (online)* | 0.74 | 1.48 | 0.69 | 0.94 | 14.69 | 8.46 | 0.84 | 0.24 | 0.08 |
| *RMIA (offline)* | 0.73 | 3.47 | 1.83 | 0.93 | **25.98** | **20.44** | 0.83 | 0.24 | 0.10 |
| *GradNorm − loss* | 0.61 | 0.01 | 0.0 | 0.83 | 0.01 | 0.0 | 0.74 | 0.0 | 0.0 |
| *GradNorm − margin* | 0.59 | 0.0 | 0.0 | 0.72 | 0.04 | 0.01 | 0.69 | 0.01 | 0.0 |
| *AdaSIF* | 0.62 | 0.01 | 0.0 | 0.87 | 0.03 | 0.01 | 0.75 | 0.0 | 0.0 |
| ***Ours*** | 0.71 | 1.48 | 0.90 | 0.90 | 9.66 | 6.73 | 0.82 | 3.67 | 2.28 |

Table T10: **Membership Inference results (full auxiliary knowledge).** Performance across all datasets under the assumption-rich setting where training hyperparameters, data distribution, and member ratio are known. Metrics reported are TPR (%) at 0.0% and 0.01% FPR, with AUC included for completeness.

| Attack | CINIC-10 AUC | @0.01 % | @0.0 % |
|---|---|---|---|
| *Attack-P* | 0.71 | 0.01 | 0.0 |
| *Attack-R* | 0.76 | 2.42 | 0.0 |
| *LiRA (online)* | **0.85** | **5.70** | **2.82** |
| *LiRA (offline)* | 0.58 | 0.41 | 0.12 |
| *RMIA (online)* | 0.83 | 0.34 | 0.06 |
| *RMIA (offline)* | 0.83 | 0.41 | 0.07 |
| ***Ours*** | 0.81 | 5.19 | 1.96 |

Table T11: **Membership Inference results (unknown fraction of members).** CINIC-10 results when the attacker does not know the proportion of training members in the candidate pool. Metrics reported are TPR (%) at 0.0% and 0.01% FPR, plus AUC.

| Attack | CIFAR-10 AUC | @0.01 % | @0.0 % | CIFAR-100 AUC | @0.01 % | @0.0 % | CINIC-10 AUC | @0.01 % | @0.0 % |
|---|---|---|---|---|---|---|---|---|---|
| *Attack-P* | 0.75 | 0.03 | 0.0 | 0.90 | 0.02 | 0.0 | 0.82 | 0.01 | 0.0 |
| *Attack-R* | 0.71 | 1.02 | 0.63 | 0.93 | 10.10 | 4.95 | 0.82 | 3.27 | 1.42 |
| *LiRA (online)* | **0.81** | 1.94 | 1.14 | **0.97** | 12.66 | 4.63 | **0.90** | **3.73** | 1.84 |
| *LiRA (offline)* | 0.53 | 0.04 | 0.01 | 0.76 | 0.0 | 0.0 | 0.70 | 0.31 | 0.17 |
| *RMIA (online)* | 0.80 | 0.34 | 0.05 | **0.97** | 10.83 | 5.54 | 0.88 | 0.28 | 0.08 |
| *RMIA (offline)* | 0.79 | 0.56 | 0.14 | **0.97** | **15.36** | **8.50** | 0.86 | 0.51 | 0.15 |
| ***Ours*** | **0.81** | **2.14** | **1.18** | 0.95 | 11.60 | 5.05 | 0.85 | 3.28 | **2.69** |

Table T12: **Membership Inference results (different data distribution).** Performance across all datasets when the candidate pool mixes in-distribution and out-of-distribution data. Metrics reported are TPR (%) at 0.0% and 0.01% FPR, with AUC included for completeness.

| Attack | CIFAR-10 AUC | @0.01 % | @0.0 % | CIFAR-100 AUC | @0.01 % | @0.0 % | CINIC-10 AUC | @0.01 % | @0.0 % |
|---|---|---|---|---|---|---|---|---|---|
| *Attack-P* | 0.59 | 0.01 | 0.0 | 0.81 | 0.01 | 0.0 | 0.71 | 0.0 | 0.0 |
| *Attack-R* | 0.67 | 1.53 | 0.49 | 0.90 | 2.97 | 1.55 | 0.79 | 1.62 | 0.37 |
| *LiRA (online)* | **0.73** | **2.56** | **1.32** | **0.92** | 13.86 | 9.93 | **0.85** | **5.81** | **3.47** |
| *LiRA (offline)* | 0.62 | 0.98 | 0.38 | 0.88 | 5.43 | 2.26 | 0.72 | 1.69 | 1.03 |
| *RMIA (online)* | 0.72 | 1.65 | 0.46 | 0.93 | 6.49 | 0.84 | 0.84 | 0.92 | 0.32 |
| *RMIA (offline)* | 0.71 | 1.89 | 0.76 | 0.92 | **15.14** | **9.99** | 0.83 | 0.79 | 0.08 |
| ***Ours*** | 0.71 | 1.48 | 0.90 | 0.90 | 9.66 | 6.73 | 0.82 | 3.67 | 2.28 |

Table T13: **Membership Inference results (unknown training configuration).** Performance across all datasets when the attacker does not know the target model's training hyperparameters. Metrics reported are TPR (%) at 0.0% and 0.01% FPR, with AUC included for completeness.

| Attack | CIFAR-10 | | | CIFAR-100 | | | CINIC-10 | | |
|---|---|---|---|---|---|---|---|---|---|
| | AUC | @0.01 % | @0.0 % | AUC | @0.01 % | @0.0 % | AUC | @0.01 % | @0.0 % |
| *LiRA (online)* | **0.80** | **0.55** | **0.17** | 0.96 | **7.90** | **2.36** | **0.88** | **2.27** | **0.66** |
| *LiRA (offline)* | 0.49 | 0.0 | 0.0 | 0.76 | 0.0 | 0.0 | 0.64 | 0.04 | 0.02 |
| *RMIA (online)* | **0.80** | 0.19 | 0.01 | **0.97** | 6.73 | 1.22 | 0.87 | 0.15 | 0.03 |
| *RMIA (offline)* | 0.79 | 0.26 | 0.0 | **0.97** | 7.35 | 1.67 | 0.86 | 0.11 | 0.02 |

Table T14: **LiRA and RMIA: offline vs. online (no auxiliary knowledge).** Results for LiRA and RMIA across all datasets in the realistic no-auxiliary-knowledge setting. The offline variant trains reference models independently of the candidate superset, while the online variant trains reference models directly on subsets of the superset, closer to the evaluation setup. Metrics reported are TPR (%) at 0.0% and 0.01% FPR, with AUC included for completeness.

| Attack | CIFAR-10 | | CIFAR-100 | | CINIC-10 | |
|---|---|---|---|---|---|---|
| | @0.01 % | @0.0 % | @0.01 % | @0.0 % | @0.01 % | @0.0 % |
| *Attack-P* | $0.02 \pm 0.0$ | $0.0 \pm 0.0$ | $0.01 \pm 0.0$ | $0.0 \pm 0.0$ | $0.01 \pm 0.0$ | $0.0 \pm 0.0$ |
| *Attack-R* | $0.23 \pm 0.10$ | $0.04 \pm 0.04$ | $0.52 \pm 0.14$ | $0.04 \pm 0.01$ | $0.31 \pm 0.19$ | $0.0 \pm 0.0$ |
| *LiRA* | $0.55 \pm 0.09$ | $0.17 \pm 0.01$ | $7.90 \pm 0.79$ | $2.36 \pm 0.30$ | $2.27 \pm 0.25$ | $0.66 \pm 0.13$ |
| *RMIA* | $0.19 \pm 0.04$ | $0.01 \pm 0.0$ | $6.73 \pm 0.84$ | $1.22 \pm 0.45$ | $0.15 \pm 0.02$ | $0.03 \pm 0.0$ |
| *GradNorm–loss* | $0.11 \pm 0.01$ | $0.01 \pm 0.0$ | $0.10 \pm 0.02$ | $0.04 \pm 0.01$ | $0.09 \pm 0.02$ | $0.01 \pm 0.0$ |
| *GradNorm–margin* | $0.02 \pm 0.01$ | $0.0 \pm 0.0$ | $0.02 \pm 0.01$ | $0.01 \pm 0.0$ | $0.03 \pm 0.0$ | $0.01 \pm 0.01$ |
| *AdaSIF* | $0.05 \pm 0.01$ | $0.0 \pm 0.0$ | $0.01 \pm 0.0$ | $0.0 \pm 0.0$ | $0.01 \pm 0.0$ | $0.0 \pm 0.0$ |
| ***ImpMIA (ours)*** | $\mathbf{2.76} \pm \mathbf{0.34}$ | $\mathbf{1.41} \pm \mathbf{0.29}$ | $\mathbf{14.86} \pm \mathbf{0.40}$ | $\mathbf{5.26} \pm \mathbf{1.01}$ | $\mathbf{5.32} \pm \mathbf{0.42}$ | $\mathbf{2.47} \pm \mathbf{0.46}$ |

Table T15: **Membership Inference Results (no auxiliary knowledge).** Performance across all datasets under the realistic no-assumptions setting. Metrics reported are TPR (%) at 0.0% and 0.01% FPR, with standard error included. This table shows the same results as in Table 1, but with standard error values reported.

| Method | Full Auxiliary-Knowledge | Unknown Training Config. | Different Distribution | Unknown Fraction of Members | NO Auxiliary-Knowledge |
|---|---|---|---|---|---|
| *Attack-R* | $4.62 \pm 0.76$ | $1.62 \pm 0.67$ | $3.27 \pm 0.53$ | $2.42 \pm 0.19$ | $0.31 \pm 0.20$ |
| *LiRA* | $7.59 \pm 0.47$ | $5.81 \pm 0.53$ | $3.73 \pm 0.70$ | $5.70 \pm 0.58$ | $2.27 \pm 0.25$ |
| *RMIA* | $0.24 \pm 0.05$ | $0.92 \pm 0.32$ | $0.28 \pm 0.15$ | $0.34 \pm 0.06$ | $0.15 \pm 0.02$ |
| **ImpMIA (ours)** | $3.67 \pm 0.95$ | $3.67 \pm 0.95$ | $3.28 \pm 0.41$ | $5.19 \pm 0.31$ | $5.32 \pm 0.42$ |

Table T16: **Assumptions influence (CINIC-10, @0.01% FPR).** Each entry is TPR (%) $\pm$ se. This table shows the same results as in Table 3, but with standard error values reported.

| Method | Full Auxiliary-Knowledge | Unknown Training Config. | Different Distribution | Unknown Fraction of Members | NO Auxiliary-Knowledge |
|---|---|---|---|---|---|
| *Attack-R* | $1.81 \pm 0.55$ | $0.37 \pm 0.26$ | $1.42 \pm 0.80$ | $0.0 \pm 0.0$ | $0.0 \pm 0.0$ |
| *LiRA* | $5.03 \pm 0.89$ | $3.47 \pm 1.01$ | $1.84 \pm 0.53$ | $2.82 \pm 0.61$ | $0.66 \pm 0.14$ |
| *RMIA* | $0.08 \pm 0.03$ | $0.32 \pm 0.09$ | $0.08 \pm 0.03$ | $0.06 \pm 0.03$ | $0.03 \pm 0.01$ |
| **ImpMIA (ours)** | $2.28 \pm 0.60$ | $2.28 \pm 0.60$ | $2.69 \pm 0.53$ | $1.96 \pm 0.38$ | $2.47 \pm 0.46$ |

Table T17: **Assumptions influence (CINIC-10, @0.0% FPR).** Each entry is TPR (%) $\pm$ se. This table shows the same results as in Table 3, but with standard error values reported.

| Method | Unknown Config. + Different Dist. | Unknown Config. + Unknown Ratio | Different Dist. + Unknown Ratio |
|---|---|---|---|
| *Attack-R* | 0.55 / 0.00 | 1.81 / 0.00 | 1.31 / 0.11 |
| *LiRA* | 2.78 / 1.31 | 3.97 / 1.78 | 2.55 / 0.44 |
| *RMIA* | 0.27 / 0.12 | 0.37 / 0.03 | 0.10 / 0.02 |
| ***ImpMIA (ours)*** | **3.28 / 2.69** | **5.19 / 1.96** | **5.32 / 2.47** |

Table T18: **Additional assumption combinations on CINIC-10.** We report TPR (%) at 0.01% / 0.0% FPR for the missing pairwise combinations of the three removed assumptions. Together with Table 3, which reports the full auxiliary-knowledge setting, each single removed assumption, and the no-auxiliary-knowledge setting, these results cover all combinations of assumptions.

# D    Binary case of the KKT equations and Implicit Bias with Weight Decay

For completeness, we also present the implicit bias of neural networks for the binary classification case, as appeared in Lyu & Li (2019); Ji & Telgarsky (2020); Haim et al. (2022), and its extension to training with weight decay that was previously considered in (Buzaglo et al., 2023).

***Implicit bias of gradient flow in homogeneous networks***: Let $\Phi(\theta; \cdot) : \mathbb{R}^d \to \mathbb{R}$ be a homogeneous ReLU network. Consider minimizing the logistic loss over a binary classification dataset $\{(x_i, y_i)\}_{i=1}^n \subseteq \mathbb{R}^d \times \{\pm 1\}$ using gradient flow. Suppose that at some time $t_0$ the network classifies all samples correctly. Then gradient flow converges in direction to a KKT point of the maximum-margin problem:

$$\min_\theta \; \tfrac{1}{2}\|\theta\|^2 \quad \text{s.t.} \quad \forall i \in [n] \; y_i \, \Phi(\theta; x_i) \geq 1.$$

The associated KKT conditions are:

$$\theta \; - \; \sum_{i=1}^n \lambda_i \, \nabla_\theta \big[ y_i \, \Phi(\theta; x_i) \big] = 0 \qquad \text{(stationarity)} \tag{4}$$

$$y_i \, \Phi(\theta; x_i) \geq 1 \qquad\qquad\qquad \text{(primal feasibility)} \tag{5}$$

$$\lambda_i \geq 0 \qquad\qquad\qquad\qquad \text{(dual feasibility)} \tag{6}$$

$$\lambda_i = 0 \quad \text{if } y_i \, \Phi(\theta; x_i) \neq 1 \qquad \text{(complementary slackness).} \tag{7}$$

***Bias with weight decay***: Previous works (Haim et al., 2022; Buzaglo et al., 2023) have further analyzed the effect of explicit weight decay in this context. For simplicity, and following Buzaglo et al. (2023), we present the analysis in the binary classification case, though the argument can be extended to the multiclass setting.

Let $\ell(\Phi(x_i; \theta), y_i)$ be a loss function that takes as input the scalar prediction of the model $\Phi(\cdot; \theta)$ on sample $x_i$, and its corresponding label $y_i$. The total regularized loss is

$$L(\theta) = \sum_{i=1}^n \ell(\Phi(x_i; \theta), y_i) \; + \; \lambda_{\text{WD}} \, \tfrac{1}{2} \|\theta\|^2.$$

Assuming convergence ($\nabla_\theta L = 0$), the parameters satisfy

$$\theta = \sum_{i=1}^n \ell_i' \, \nabla_\theta \Phi(x_i; \theta), \qquad \ell_i' = -\frac{1}{\lambda_{\text{WD}}} \frac{\partial \, \ell(\Phi(x_i; \theta), y_i)}{\partial \Phi(x_i; \theta)}. \tag{8}$$

This relation shows that the trained weights again lie in the span of per-sample gradients, with coefficients $\{\ell_i'\}$ determined by the derivative of the loss. Importantly, equation 8 is structurally equivalent to the stationarity condition in the KKT system of the max-margin problem: in both formulations, the parameters are expressed as a linear combination of margin-gradient directions. Furthermore, if $\ell$ is the logistic loss function, then the coefficients $\{\ell_i'\}$ of this linear combination are non-negative.

# E   Limitations of Average-Case Metrics

A common evaluation practice in the membership inference literature is to report average-case metrics such as balanced accuracy or ROC-AUC. While convenient, these metrics are misaligned with the privacy risks that matter in practice.

First, average-case metrics obscure worst-case behavior. Balanced accuracy treats false positives and false negatives symmetrically, yet in privacy attacks the costs are asymmetric: false positives (incorrectly labeling non-members as members) undermine reliability, whereas false negatives are typically less harmful. Second, aggregate metrics such as AUC average performance across the entire ROC curve, including regions of high false-positive rates that are irrelevant in practice. As emphasized by Carlini et al. (2022), an attack may achieve high AUC while completely failing to recover any members at $\leq 0.1\%$ FPR. Conversely, an attack that reliably recovers only a small subset of members may achieve modest AUC but still constitute a severe privacy breach. For this reason, recent work on membership inference (Carlini et al., 2022; Zarifzadeh et al., 2023) has adopted TPR at low FPR as the primary evaluation metric.

This mismatch is evident in our results. Table 1 reports performance of several attacks in the No-Auxiliary-Knowledge setting. On CIFAR-10, both our attack and the GradNorm baseline achieve AUC $\approx 0.81$. However, this similarity is misleading: GradNorm leverages model confidence, and its apparent effectiveness comes from the fact that non-members in this scenario have low confidence. By contrast, as shown by Haim et al. (2022), the truly memorized samples are those near the decision boundary. Our attack explicitly targets such near-margin points, yielding much higher TPR at very low FPR (e.g., 2.76% vs. 0.11% at 0.01% FPR). This gap illustrates why average-case metrics like AUC are problematic: they make GradNorm appear competitive, while in reality it fails where privacy evaluation matters most, the low-FPR regime

# F   Broader Impact Concerns

Membership inference attacks are important tools for evaluating whether trained models unintentionally reveal information about their training data. ImpMIA can support privacy auditing by helping model developers identify such leakage, evaluate privacy defenses, and make informed decisions regarding model deployment and the release of model parameters.

However, membership inference can also be used maliciously to determine whether specific samples, potentially containing sensitive or personal information, were used for training. For example, an attacker could try to infer whether an individual's medical record was included in a healthcare model's training set, whether a person's image appeared in a face-recognition dataset, or whether private user data were used to train a commercial model. This risk is shared with prior work in the field: tools developed to expose and defend against privacy attacks may also facilitate such attacks. Our findings further highlight the importance of limiting access to model parameters when models are trained on sensitive data and of developing effective privacy-preserving training and model-release mechanisms.

