# OpenReview forum: "ImpMIA: Leveraging Implicit Bias for Membership Inference Attack"
_TMLR — Decision pending for TMLR_

### Review · Reviewer_rbSG · 2026-05-03

**Summary Of Contributions:**

This work analyzes both theoretically and empirically the use of the Karush–Kuhn–Tucker (KKT) condition for membership inference attacks (MIA) in classification tasks. Intuitively speaking, the KKT condition mean the final model weights are a convex combination of the gradients of the training data, which the authors use to identify which elements out of a mixed set of training and fresh data elements were actually used to train the model (MIA). Compared to other methods, this technique requires access to the model weights which is a stronger assumption relative to the API access required by some other methods, but on the other hand it saves the need for auxilery information such as the exact training method, access to the underlying distribution, etc. Previous works used the KKT condition to reconstruct relatively small training datasets from the final model, but the use of this technique for MIA is novel (though, was theoretically discussed in the past, in a work that was not cited [1]).

The presentation is clear, rigorous and accessible, the theoretical background as well as the empirical approach are well defined and easy to follow, and the appendices contain most of the additional information required to fully understand the work. The empirical results are quite impressive. This method approximately recovers the SotA results under the auxilery information assumptions (e.g., LiRA), and significantly supersedes them when removing these assumptions (though, as mentioned, it does requires access to the model weights). The only significant weakness I've found is in the authors framing of "black-box" and "white-box" settings, which tie together the question of model access (only in white-box) and the auxilery assumptions (only in black-box), as I discuss below.

Overall, this is a great work which I am happy I got the chance to read, and I strongly recommend to accept it.

[1] Smorodinsky, G., Vardi, G., & Safran, I. (2024). Provable Privacy Attacks on Trained Shallow Neural Networks. arXiv preprint arXiv:2410.07632.

**Audience:**

Yes

**Audience Explanation:**

I expect many readers to find this contribution useful. The assumptions it uses are quite reasonable, the authors justify their implementation decisions (up to some minor details I mention in the next section), they provide sufficient implementation details for follow-up work, and the results can be used for various purposes including privacy auditing.

**Claims And Evidence:**

Yes

**Claims Explanation:**

The theoretical background is well establish, the experiments are well explained, their conclusions support the claims it makes, and most implementation choices are well justified. The choice made by the authors to focus on the extreme low end of FPR significantly contributes to the validity of their results, especially if MIA is used for privacy auditing.

However, there are a few problems I think should be addressed.
1. The main issue is the framing of white-box vs. black-box settings. The authors tie together two questions that in my mind are separate from each other. The first is the level of access the attacker has to the trained model - API, final model, or all gradients, and the second is the amount of auxilery information the attacker has, most notably - training hyperparameters, access to additional points from the underlying distribution, and fraction of training data in the evaluation set. The authors frame the MIA task as falling into one of two possible categories, white-box - where the attacker has access to the final model or all gradients but no auxilery information, or black-box - where the attacker has only API access but also to the auxilery information, but in fact - these two questions are actually independent, and should be presented as a 2 by 2 set of options. In my opinion, this will not weaken the importance of the results, while avoiding possible confusing or misleading the reader.
2. The distinction between the online and offline versions of LiRA was not made clear enough, which seems to affect the the magnitude of the relative improvement of this method. Is the offline version of LiRA in Fig. 3 not suitable for the No-Auxiliary-Knowledge setting, if so - why, and if not - why was the offline version not compared to?
3. This method holds for classification tasks (or, explain how it may be extended).

**Requested Changes:**

Critical:
1. Clarify the three issues raised in the previous section.
2. Cite [1] who already analyzed theoretically performing MIA using KKT for some specific setting (shallow ReLU NNs).

Recommendations:
1. Section 4.2 can benefit from some additional implementation details that were differed to Appendix A. Even if just by adding a single paragraph explaining the steps (e.g., first paragraph of Appendix A), which distills the choices made in the implementation.
2. The implementation is "KKT-inspired" more than a KKT-implementation, with the difference consisting of many decisions, some better motivated than others. It seems like some of them follow the choices made by previous works, while others are novels decisions. I think the reader will greatly benefit from making this distinction more explicit, and adding more details for the new ones (e.g., centering and normalizing were not clearly defined, but I assume this is a common operation used in previous works).
3. I found Table 3 very useful in assessing how each assumption affects the MIA results. I think the work will greatly benefit from additional comparisons in the appendix, including (a) an extended table containing all 2^3 combinations of assumptions, (b) the ROCs corresponding to the assumptions compared in the table(s), and (c) treating the assumptions as a scale rather than a binary property and comparing the effect of the magnitude of the change (e.g., the magnitude of the change in the training configuration).
4. While the authors provide a detailed explanation of the theoretical foundation of this work, it might benefit from explicitly pointing out that the KKT condition amounts to the model laying in the subspace spanned by the gradients of the training dataset, and the MIA attack amounts to identifying the elements spanning this subspace. While this insight is implied by all explanations, I believe some readers will benefit from explicitly stating this intuition.
5. I was somewhat confused what is the source of the CINIC-10 OOD dataset, if CINIC-10 itself was used as the OOD dataset for CIFAR-10. A short clarification will be very helpful.

Nitpicking:
1. To the best of my understanding, "gradient descent with a small step size" is not equivalent to gradient flow. One can either say that it well approximates it, or that it is equal for vanishingly small step size.
2. The term "superset" implies it contains the entire training-set, while this work extends the results even to the case where only a subset of the training data is contained in the evaluation set.

---

> ### Author Response · Authors · 2026-06-12
> **Official Comment by Authors**
>
> **W1: Better framing of the attacker’s level of access to the trained model (white-box vs. black-box) and the amount of auxiliary information available to the attacker.**
> The main issue is the framing of white-box vs. black-box settings. The authors tie together two questions that in my mind are separate from each other. The first is the level of access the attacker has to the trained model - API, final model, or all gradients, and the second is the amount of auxilery information the attacker has, most notably - training hyperparameters, access to additional points from the underlying distribution, and fraction of training data in the evaluation set. The authors frame the MIA task as falling into one of two possible categories, white-box - where the attacker has access to the final model or all gradients but no auxilery information, or black-box - where the attacker has only API access but also to the auxilery information, but in fact - these two questions are actually independent, and should be presented as a 2 by 2 set of options. In my opinion, this will not weaken the importance of the results, while avoiding possible confusing or misleading the reader.
>
> **Response:**
> We have revised the introduction to better frame and explain those aspects.
>
> ---
>
> **W2: The distinction between the online and offline versions of LiRA was not made clear enough, which seems to affect the magnitude of the relative improvement of this method.** Is the offline version of LiRA in Fig. 3 not suitable for the No-Auxiliary-Knowledge setting, if so - why, and if not - why was the offline version not compared to?
>
> **Response:**
>
> In the paper that introduced LiRA, the authors present two versions of their attack: “online” and “offline”. The “online” version is called online because it trains reference models after seeing the candidate sample whose membership is being evaluated, while the “offline” version trains the reference models before seeing the sample, or set of samples, for evaluation.
> More specifically, the online version aims to determine each sample’s membership by training one set of reference models in which this sample is included in the training data, and another set in which it is not. Then, given the attacked target model, it checks whether the sample’s behavior under the target model is closer to the distribution of reference models trained with this sample or to those trained without it. In contrast, the offline version trains the reference models before having access to the attacked sample set, using a different set of samples.
> Both versions can be used in our set-based setting, and both versions still strongly depend on auxiliary knowledge, since both train reference models that aim to mimic the target model’s behavior. Therefore, both versions can be tested in all our scenarios, including the No-Auxiliary-Knowledge setting. We now clarify this distinction in Appendix C.1.
> Importantly, we already include a comparison of the online and offline versions of LiRA in Table T14 in the appendix, which shows that the online version provides better results. For clarity in the main paper, we chose to present only the online version, which is the stronger variant.
>
> ---
>
> **W3: This method holds for classification tasks (or, explain how it may be extended).**
>
> **Response:**
> Indeed, both the underlying theory and our current work focus on classification tasks, which constitute a broad and well-studied area in general and in membership inference specifically.

---

> ### Author Response · Authors · 2026-06-12
>
> ***Requested Changes***
>
> **C1: Clarify the three issues raised in the previous section.**
>
> **Response:**
> See responses above.
>
> ---
>
> **C2: Cite [1] who already analyzed theoretically performing MIA using KKT for some specific setting (shallow ReLU NNs).**
>
> **Response:**
> We thank the reviewer for the pointer. We added the citation and clarified its relation to our work.
>
> ---
> ---
>
> **R1: Section 4.2 can benefit from some additional implementation details that were differed to Appendix A.** Even if just by adding a single paragraph explaining the steps (e.g., first paragraph of Appendix A), which distills the choices made in the implementation.
>
> **Response:**
> We added a compact implementation summary at the beginning of Section 4.2, outlining the practical pipeline from candidate filtering and augmentation to block-wise gradient construction, coefficient optimization, aggregation, and margin-based post-processing. The full implementation details remain in Appendix A.
>
> ---
>
> **R2: The implementation is "KKT-inspired" more than a KKT-implementation, with the difference consisting of many decisions, some better motivated than others. It seems like some of them follow the choices made by previous works, while others are novels decisions.** I think the reader will greatly benefit from making this distinction more explicit, and adding more details for the new ones (e.g., centering and normalizing were not clearly defined, but I assume this is a common operation used in previous works).
>
> **Response:**
> We agree that our implementation is KKT-inspired rather than a direct implementation of the exact KKT system. We have clarified this throughout the paper and revised Appendix A to make the distinction explicit. Specifically, the KKT-derived core consists of using margin gradients and optimizing their coefficients, while block-wise optimization, layer/filter grouping, centering, normalization, cosine-similarity optimization, and coefficient debiasing are practical stabilization steps introduced to scale the method to larger networks and candidate pools.
>
> ---
>
>
> **R3:  Extend Table 3 for more scenarios:** I found Table 3 very useful in assessing how each assumption affects the MIA results. I think the work will greatly benefit from additional comparisons in the appendix, including (a) an extended table containing all 2^3 combinations of assumptions, (b) the ROCs corresponding to the assumptions compared in the table(s), and (c) treating the assumptions as a scale rather than a binary property and comparing the effect of the magnitude of the change (e.g., the magnitude of the change in the training configuration).
>
> **Response:**
> (a) In the revised appendix, we added the missing pairwise combinations of the three removed assumptions (Table T18). Together with Table 3, this covers all 8 combinations. Each entry reports TPR (%) at 0.01% / 0.0% FPR.
>
> | Method | Unknown Config. + Different Dist. | Unknown Config. + Unknown Ratio | Different Dist. + Unknown Ratio |
> |:---|---:|---:|---:|
> | Attack-R | 0.55 / 0.00 | 1.81 / 0.00 | 1.31 / 0.11 |
> | LiRA | 2.78 / 1.31 | 3.97 / 1.78 | 2.55 / 0.44 |
> | RMIA | 0.27 / 0.12 | 0.37 / 0.03 | 0.10 / 0.02 |
> | ImpMIA (ours) | 3.28 / 2.69 | 5.19 / 1.96 | 5.32 / 2.47 |
>
> (b) We also added ROC curves for the scenarios reported in Table 3, so the comparison is not limited to the low-FPR numbers in the table (see Fig. S2).
> (c) Regarding gradual changes in the assumptions, we agree this is useful, but due to the high cost of rerunning all reference-model baselines, we could not include a full sweep in this revision.
>
>
> ---
>
>
> **R4: While the authors provide a detailed explanation of the theoretical foundation of this work, it might benefit from explicitly pointing out that the KKT condition amounts to the model laying in the subspace spanned by the gradients of the training dataset, and the MIA attack amounts to identifying the elements spanning this subspace. While this insight is implied by all explanations, I believe some readers will benefit from explicitly stating this intuition.**
>
> **Response:**
> We thank the reviewer for this suggestion. We now state this intuition explicitly in Section 4.1: the KKT condition places the trained parameter vector in the span of the training-sample margin gradients, while ImpMIA aims to identify the candidate gradients that best reconstruct this parameter vector.

---

> ### Author Response · Authors · 2026-06-12
>
> **R5: Clarify the source of the CINIC-10 OOD dataset.**
>
> **Response:**
> We thank the reviewer for pointing out this ambiguity. We have clarified the OOD construction in the revised text (see Section 5.2). CINIC-10 is used as an OOD source only for the CIFAR-10 experiments, specifically using its ImageNet-derived samples. When CINIC-10 itself is used as the in-distribution dataset, the OOD samples are instead taken from OpenImages.
>
> ---
>
>
> **R6: To the best of my understanding, "gradient descent with a small step size" is not equivalent to gradient flow. One can either say that it well approximates it, or that it is equal for vanishingly small step size.**
>
> **Response:**
> Indeed, gradient descent with a finite step size is not exactly equivalent to gradient flow. Gradient flow corresponds to the continuous-time limit of gradient descent as the step size tends to zero, while gradient descent with a sufficiently small step size can approximate it. We have corrected this statement in the revised text.
>
> ---
>
>
> **R7: The term "superset" implies it contains the entire training-set, while this work extends the results even to the case where only a subset of the training data is contained in the evaluation set**
>
> **Response:**
> We agree and have replaced the term “superset” with “candidate set”, which more accurately reflects that the evaluation set may contain only a subset of the training data.

---

### Review · Reviewer_YEN4 · 2026-05-13

**Summary Of Contributions:**

The paper studies a *set-based No-Auxiliary-Knowledge* setting of Membership Inference Attacks (MIA): given a super set candidate pool $X_{sup}$ s.t. the unknown $X_{train} \subset X_{sup}$, the adversary assigns a real-valued *membership* score to each $x_i \in X_{\rm sup}$, where high score means that the adversary assings $x_i \in X_{\rm train}$ to be a member. The adversary is also given the trained parameters $\theta$, i.e. a white-box MIA setting. To solve this problem, the paper proposes the ImpMIA attack, building on the *intuition* from the implicit bias/maximum-margin formulation of neural networks. At a high level, the attack considers that the trained vector parameter of a neural net, trained on gradient descent, can be thought of as $\theta \approx \sum_{i \in X_{train}} \lambda_i g_i$, where $g_i$ is the (margin) gradient of a training sample, and $\lambda_i$ are the "optimised" KKT multipliers. Since the attacker does not know $X_{train}$, ImpMIA computes these gradients for all candidates $x_i\in X_{sup}$, stacks them into $A=[g_1,\dots,g_M]$, and optimizes coefficients $\lambda \in \mathbb{R}^M$ so that $A \lambda \approx \theta$. The resulting coefficient $\lambda_i$, after blockwise optimisation, normalisation, aggregation over blocks/augmentations, and post-processing, is used as the membership score of sample $x_i$. The intuition is that true members should contribute more consistently to reconstructing the trained weights, and hence receive larger coefficients. Empirically, the paper reports that ImpMIA outperforms both black-box reference-model attacks and existing white-box attacks in the proposed no-auxiliary-knowledge setting, especially at very low false-positive rates. The experimental section includes results on CIFAR-10, CIFAR-100, and CINIC-10, ablations on the removed assumptions, architecture variants, partial training-set coverage, score aggregation, weight decay, pre-filtering, and running time.

## Strengths:
- **Original and interesting idea**: The paper proposes a clever use of implicit-bias/KKT conditions for membership inference. The connection between maximum-margin and membership scoring is appealing and, to my knowledge, not explored in the MIA literature.
- **Fewer auxiliary assumptions than reference-model attacks**: A major strength is that ImpMIA does not require training reference/shadow models, and therefore avoids several assumptions used by strong black-box attacks, such as knowing the target training configuration, matching the non-member distribution, or knowing the member ratio in the inference pool. Although ImpMIA is still nontrivial to run, avoiding the training of hundreds of reference models is also a meaningful computational advantage.
- **Strong empirical performance in the intended setting.** The reported gains at very low FPR are compelling, especially because low-FPR performance is the most relevant regime for privacy auditing. The comparisons include both black-box and white-box baselines, and the results suggest that the proposed signal is useful.


## Weaknesses:
- **The method remains heuristic/theory-inspired rather than theoretically justified for the actual experimental setting**: The motivating implicit-bias results hold under idealised assumptions such as homogeneous networks, separability, are asymptotic, and for gradient flow (or discrete SGD but with additional stronger assumptions). The experiments, however, use practical architectures and training procedures such as ResNets, SGD, augmentations, and weight decay.
- **The superset assumption is still quite strong**: ImpMIA requires the attacker to possess a candidate pool that contains the training data, or at least a non-negligible fraction of it. This is a meaningful restriction.
- **The $\lambda$ optimisation pipeline is complex and under-analysed**: The final score depends not only on the KKT-inspired objective, but also on filtering, horizontal flips, blockwise optimisation, centring/normalisation, aggregation, margin-based boosting, and post-processing. The ablations help, but it is still hard to disentangle exactly which components are essential and how robust the method is to these design choices.

## Questions:
- **Why is the full $\lambda$ coefficient optimisation necessary compared to simpler scalar product scores?**: Since the attack's starting point is thinking of $\theta \approx \sum_{i \in X_{train}} \lambda_i g_i$, why not take as a score for some sample $x_i$ the scalar product (or correlation etc) of $g_i$ with $\theta$? This would alleviate the need for $X_{sup}$ and the full optimisation pipeline. Did the authors compare against such simple scalar product scores? If yes, how much does the global optimisation over all $\lambda_i$'s improve over these simpler alternatives?

- **Why not impose sparsity/L1 regularisation on the $\lambda$ optimisation?**:  The KKT motivation suggests that at most $n_{train}$ out of the $M$ coefficients should be non-zero, so why not add explicit sparsity/L1 regularisation in the optimisation step?

**Audience:**

Yes

**Audience Explanation:**

I think the paper would be of clear interest to at least part of the TMLR audience, especially researchers working on membership inference and privacy auditing. The central idea is original and appealing. Even if the final method is ultimately heuristic in practical deep-learning settings, it gives a fresh way to think about how training samples may be encoded in model parameters.

**Broader Impact Concerns:**

The paper does not appear to include a dedicated Broader Impact statement. Since the work proposes a new membership inference attack, this would be a useful and appropriate addition.

**Claims And Evidence:**

Yes

**Claims Explanation:**

The main empirical claims are supported by reasonably clear and convincing evidence. The paper evaluates ImpMIA against strong black-box and white-box MIA baselines across several datasets, and reports consistent gains in the no-auxiliary-knowledge setting, especially at very low false-positive rates, which is the most relevant regime for privacy auditing. The authors also isolate the effect of removing each auxiliary assumption used by reference-model attacks, which helps support their central claim that ImpMIA is more robust when the attacker does not know the target training configuration, data distribution, or member ratio. The experimental section is extensive and includes architecture variants, ablations of the scoring/aggregation pipeline, partial training-set coverage, weight decay, pre-filtering, and runtime analysis.

That said, the theoretical evidence is less complete than the empirical evidence. The implicit-bias motivation is mathematically plausible and well explained, but the actual attack is applied in practical settings where the formal assumptions behind the implicit-bias/KKT theory do not fully hold. Thus, I view the theory mainly as a compelling source of intuition rather than a complete justification of the method.

**Requested Changes:**

- **Provide clean pseudo-code for the complete attack.**
  The current description of ImpMIA is understandable at a high level, but the full implementation involves several steps. I strongly recommend adding a single self-contained pseudo-code box with clear inputs, outputs, and all the steps in order.

- **Reposition the introduction more clearly within the white-box MIA literature.**
  The current introduction puts a lot of emphasis on the assumptions of black-box reference-model attacks. While this is relevant, ImpMIA is  a white-box attack, and I think the paper would be stronger if it were positioned more directly against prior white-box MIAs. In particular, the introduction should better explain what existing white-box attacks use as their signal, why these signals may be insufficient, and how ImpMIA provides a new kind of white-box signal based on implicit-bias/KKT structure.

- **Be more explicit about the heuristic nature of the attack.**
  The implicit-bias theory provides a compelling motivation, but the practical attack is applied to architectures/training regimes that do not exactly satisfy the formal assumptions behind the theory. I recommend clearly stating that the method is theory-inspired rather than theoretically guaranteed in the practical ResNet/SGD/augmentation/weight-decay setting. This would avoid overclaiming and make the paper’s contribution more precise.

- **Add a clearer computational analysis.**
The paper reports runtime, but it would benefit from a clearer complexity analysis in terms of the number of candidates \(M\), number of parameters \(p\), number of blocks, memory requirements, and optimization iterations. Since the attack involves per-sample gradients and large linear systems/optimisations, scalability is an important practical question.

- **Minor: Define all loss terms and scoring quantities before use.**
  Some quantities, such as $L_{neg}$ and $L_{marg}$, appear without being defined at all in the paper. Please make sure every objective term, regularizer, and scoring component is explicitly introduced before it is used.

- **Optional but valuable: add a synthetic ablation isolating the KKT assumptions.**
  This is not essential, but it would significantly strengthen the paper. Since the attack is motivated by implicit-bias/KKT theory, it would be useful to include a controlled synthetic experiment where the assumptions are gradually varied: homogeneous vs. non-homogeneous networks, separable vs. non-separable data, small learning rate approximating gradient flow vs. standard learning rate, very long training vs. finite-time training, with/without weight decay, etc. Such an experiment would clarify how much of ImpMIA’s performance is genuinely tied to the KKT/implicit-bias theory assumptions.

---

> ### Author Response · Authors · 2026-06-12
> **Official Comment by Authors**
>
> We thank the reviewer for the time and effort devoted to evaluating our paper and for the constructive suggestions. To address the reviewer’s comments, we conducted additional experiments and provide a detailed response to each point below. The new experimental results and the corresponding revisions have been incorporated into the updated manuscript, with all changes highlighted in blue.
>
> **W1. The method remains heuristic/theory-inspired rather than theoretically justified for the actual experimental setting:** The motivating implicit-bias results hold under idealized assumptions such as homogeneous networks and separability, are asymptotic, and apply to gradient flow—or to discrete SGD under additional stronger assumptions. The experiments, however, use practical architectures and training procedures such as ResNets, SGD, augmentations, and weight decay.
>
> **Response:** We agree. As suggested, we have clarified this limitation in the paper and added experiments and discussion examining the connection between the theory and our practical setting (see our responses below).
>
> ---
>
> **W2. The superset assumption is still quite strong:** ImpMIA requires the attacker to possess a candidate pool that contains the training data, or at least a non-negligible fraction of it. This is a meaningful restriction.
>
> **Response:**
> Indeed, the superset assumption is meaningful and does not hold in every scenario. At the same time, it provides an efficient and effective way to harness implicit bias without being limited by other strong assumptions commonly made by existing methods. Importantly, high coverage is plausible in many practical settings: an attacker may have access to large candidate pools, such as all publicly available images in a given domain, and our method’s efficiency allows it to scale to such pools without training reference models. Moreover, as shown in Appendix B.3, even with low coverage (only 10%), our results are comparable to the strongest competitors, while with higher coverage our method clearly surpasses them (Table 1 in the paper).
>
> ---
>
> **W3. The $\lambda$ optimization pipeline is complex and under-analyzed:** The final score depends not only on the KKT-inspired objective, but also on filtering, horizontal flips, block-wise optimization, centering/normalization, aggregation, margin-based boosting, and post-processing. The ablations help, but it is still hard to disentangle exactly which components are essential and how robust the method is to these design choices.
>
> **Response:**
> As acknowledged by the reviewer, we provide several ablations that, while not covering every component, give a strong indication of which parts are most important, as well as the importance of our proposed optimization and its dependence on different assumptions.

---

> ### Author Response · Authors · 2026-06-12
>
> **Q1. Why is the full $\lambda$ coefficient optimization necessary compared to simpler scalar-product scores?**
> Since the attack's starting point is the relation $\theta \approx \sum_{i \in X_{\mathrm{train}}} \lambda_i g_i$, why not use the scalar product, correlation, or another similarity measure between $g_i$ and $\theta$ as the score for a sample $x_i$? This would alleviate the need for $X_{\mathrm{sup}}$ and the full optimization pipeline. Did the authors compare against such simple scalar-product scores? If so, how much does the global optimization over all $\lambda_i$ improve over these simpler alternatives?
>
> **Response:**
> We agree that scalar-product scores are a natural baseline. However, the KKT condition motivates a global reconstruction problem rather than an independent per-sample alignment score: the coefficients $\lambda_i$ measure how much each candidate contributes to jointly reconstructing $\theta$ with all other candidates. A scalar product $\langle g_i,\theta\rangle$ ignores this and therefore does not estimate the KKT multiplier.
> This can already be seen in the homogeneous setting. Let $m_i(\theta)=\Phi_{y_i}(\theta;x_i)-\max_{j\neq y_i}\Phi_j(\theta;x_i)$ be the margin of sample $i$, and let $g_i=\nabla_\theta m_i(\theta)$. For an $L$-homogeneous network, Euler’s theorem gives
> $$
> \langle g_i,\theta\rangle = \langle \nabla_\theta m_i(\theta),\theta\rangle = Lm_i(\theta).
> $$
> Thus, the scalar-product score collapses to a margin/confidence score up to a constant factor, rather than recovering $\lambda_i$. This motivates the full coefficient optimization.
> We also evaluated a normalized scalar-product baseline, using cosine similarity, on CIFAR-10 under the No-Auxiliary-Knowledge setting, using the same pre-filtering and augmentations as ImpMIA, but without optimizing the $\lambda$ coefficients. Each candidate is scored independently by the cosine similarity between its margin gradient $g_i$ and the trained weights $\theta$.
>
> | Method | AUC | TPR @ 0.01% FPR | TPR @ 0.0% FPR |
> |:---|---:|---:|---:|
> | Cosine similarity baseline | 0.71 | 0.03% | 0.0% |
> | ImpMIA (ours) | 0.81 | 2.76% | 1.41% |
>
> These results show that the independent cosine-similarity score is much weaker, supporting the need for the global $\lambda$ optimization.
>
>
>
> ---
>
> **Q2. Why not impose sparsity or L1 regularization on the $\lambda$ optimization?**
> The KKT motivation suggests that at most $n_{\mathrm{train}}$ out of the $M$ coefficients should be non-zero. Why not add explicit sparsity or L1 regularization to the optimization step?
>
> **Response:**
>
>
> We agree that sparsity is a natural regularizer in our setting. In fact, our implementation already uses a variant of $L_1$ regularization: a margin-dependent weighted $L_1$ penalty of the form $\sum_i |\lambda_i|/d_i$, where $d_i$ is the distance of sample $i$ from the decision boundary. The reason for using a margin-dependent penalty is that, according to the KKT equations, influential training samples are expected to lie near the decision boundary and receive large coefficients. However, only a sparse subset of near-boundary candidates is expected to receive large $\lambda$ values. The margin-dependent regularizer is therefore designed to encourage sparsity mainly in this margin-relevant region, rather than uniformly across all candidates. We now clarify this regularization choice in Appendix A.

---

> ### Author Response · Authors · 2026-06-12
>
> ***Requested Changes***
>
> **RC1: Provide clean pseudo-code for the complete attack.**
>
> **Response:**
> Pseudo-code for the complete attack has been added to Appendix A.
>
> ---
>
> **RC2: Reposition the introduction more clearly within the white-box MIA literature.**
> The current introduction puts a lot of emphasis on the assumptions of black-box reference-model attacks. While this is relevant, ImpMIA is a white-box attack, and I think the paper would be stronger if it were positioned more directly against prior white-box MIAs. In particular, the introduction should better explain what existing white-box attacks use as their signal, why these signals may be insufficient, and how ImpMIA provides a new kind of white-box signal based on implicit-bias/KKT structure.
>
> **Response:**
> Following the reviewer’s suggestion, we have revised the introduction to address these points more clearly and to better position ImpMIA within the white-box MIA literature.
>
> ---
>
> **RC3: Be more explicit about the heuristic nature of the attack.**
> The implicit-bias theory provides a compelling motivation, but the practical attack is applied to architectures/training regimes that do not exactly satisfy the formal assumptions behind the theory. I recommend clearly stating that the method is theory-inspired rather than theoretically guaranteed in the practical ResNet/SGD/augmentation/weight-decay setting. This would avoid overclaiming and make the paper’s contribution more precise.
>
> **Response:**
> As suggested, we have clarified this point both in the introduction and in the method section.
>
> ---
>
> **RC4: Add a clearer computational analysis.**
> The paper reports runtime, but it would benefit from a clearer complexity analysis in terms of the number of candidates ($M$), number of parameters ($p$), number of blocks, memory requirements, and optimization iterations. Since the attack involves per-sample gradients and large linear systems/optimizations, scalability is an important practical question.
>
> **Response:**
> We have now significantly extended Appendix B.11, “Running Time”, to include a more detailed analysis of computational complexity and memory requirements.
>
> ---
>
>
> **RC5: Minor: Define all loss terms and scoring quantities before use.**
> Some quantities, such as $L_{\mathrm{neg}}$ and $L_{\mathrm{marg}}$, appear without being defined in the paper. Please ensure that every objective term, regularizer, and scoring component is explicitly introduced before it is used.
>
> **Response:**
> We have revised Appendix A to explicitly define the optimization terms when introducing the objective. In particular, we now define $L_{\mathrm{neg}}$, $L_{\mathrm{marg}}$, and the regularization weights $\alpha$ and $\beta$, and clarify the scoring and post-processing components used in the pipeline.
>
> ---
>
>
> **RC6: Optional but valuable: Add a synthetic ablation isolating the KKT assumptions.**
> This is not essential, but it would significantly strengthen the paper. Since the attack is motivated by implicit-bias/KKT theory, it would be useful to include a controlled synthetic experiment in which the assumptions are gradually varied: homogeneous vs. non-homogeneous networks, separable vs. non-separable data, a small learning rate approximating gradient flow vs. a standard learning rate, very long vs. finite-time training, and training with vs. without weight decay. Such an experiment would clarify how much of ImpMIA’s performance is genuinely tied to the assumptions underlying the KKT/implicit-bias theory.
>
> **Response:**
> We have added an ablation on the number of training epochs, which directly tests how the attack behaves as training moves closer to the long-time regime assumed by the theory.
> | Training epochs | AUC | TPR @ 0.01% FPR | TPR @ 0.0% FPR |
> |:---|---:|---:|---:|
> | 20 | 0.65 | 0.08% | 0.04% |
> | 100 | 0.78 | 1.46% | 0.87% |
> | 400 | 0.81 | 2.76% | 1.41% |
> | 1000 | 0.81 | 3.01% | 1.55% |
>
> The results show that ImpMIA improves with longer training: AUC increases from 0.65 at 20 epochs to 0.81 at 400 epochs, while TPR at 0.0% FPR increases from 0.04% to 1.41%. This supports the connection between our KKT-inspired signal and the long-time implicit-bias regime. We added this ablation to the appendix (App. B.8). In addition, the paper already includes an ablation comparing training with and without weight decay.
>
> ---
> ---
>
>
> **Broader Impact Concerns: A Broader Impact section should be added to list applications where these may be used for malicious purposes.**
>
> **Response:**
> We agree and have added a Broader Impact Concerns section to the revised paper (see Section F in the appendix).

---

### Review · Reviewer_ZaLM · 2026-06-01

**Summary Of Contributions:**

The paper proposes a white-box membership attack for models under the assumptions that we know the weights of the model being attacked and are given a superset of its training data. The method exploits the max-margin implicit bias of neural networks and uses this to develop a linear system that represents model parameters as linear combinations of gradients of datapoints, with the insight that the weights of this linear combination should be higher for training examples as opposed to other datapoints. This method is empirically tested against existing black-box and white-box membership attacks and shows large improvements of TPR at low FPR. Importantly, a systematic analysis shows how various black-box attacks rely heavily on assumptions that training configurations, data splits, and data distribution are known. Removing each one shows that these black-box attacks can fail catastrophically. This method is clean and intuitive, does not involve training any reference models, and results are strong. There are a few minor claims that should be explained/validated further (see below), but other than that, it seems like a well-supported idea and paper.

**Audience:**

Yes

**Audience Explanation:**

Performing efficient membership inference attacks under minimal assumptions is an important problem and this paper proposes a nice simple idea in this domain, so it should be valuable to the audience.

**Broader Impact Concerns:**

Probably a Broader Impact section should be added to list applications where these may be used for malicious purposes.

**Claims And Evidence:**

No

**Claims Explanation:**

1. In Section 4.2 it is written that the lambda is optimized block-wise across parameters. What are the implicit assumptions that allow us to do this? I understand that this may be done for computational reasons which is fair, but I am not quite sure about the extra claim that this improves conditioning. Can you experimentally validate this claim that block-wise optimization improves conditioning?
2. My main concern about the settings chosen for Section 5.2 is with the change in training configuration for the reference model. I notice that learning rate is dropped as well as the number of epochs is dropped. With a LR drop, I would imagine convergence is slower, so is it fair to jointly drop the number of epochs by 1/4? I worry that the reference model in these attacks is just not trained to convergence which would explain the large drop in performance via training configuration. How can we decouple the "knowledge of training configuration" vs "strength of reference model" to more strongly back up the claim?

**Requested Changes:**

My main two changes (critical for acceptance) are regarding the two claims above:
1) Justification of block-wise optimization of lambda
2) Justification of the precise training configuration change for black-box attacks.

Strengthen the work:
1) It would be helpful to try this attack on one different architecture to see if these claims are specific to this ResNet model. Alternatively, a discussion in the appendix of which models have been studied in prior work to have a max-margin bias may also be helpful, as this would guide for which architectures one might expect this method to work.

---

> ### Author Response · Authors · 2026-06-12
> **Official Comment by Authors**
>
> We thank the reviewer for the time and effort devoted to evaluating our paper and for the constructive suggestions. To address the reviewer’s comments, we conducted additional experiments and we provide a detailed response to each point below. The new experimental results and the corresponding revisions have been incorporated into the updated manuscript, with all changes highlighted in blue.
>
>
> **1. Justification of block-wise optimization of $\lambda$ :** (a) What are the implicit assumptions that allow $\lambda$ to be optimized block-wise across parameters? (b) Can you experimentally validate the claim that block-wise optimization improves conditioning?
>
> **Response:**
> **(a)** The KKT equations hold for any subset of the model parameters, with each parameter corresponding to an equation. In our formulation, each optimization block contains a subset of these equations, evaluated over all candidate samples. Therefore, optimizing $\lambda$ block-wise does not require an additional independence assumption between parameter groups. Each block corresponds to a valid subset of the original KKT system.
>
> **(b)** Different layers of a neural network can have substantially different parameter and gradient statistics. In our preliminary experiments, performing a single gradient-based optimization using equations sampled across all parameters resulted in unstable optimization and substantially weaker attack performance. At the same time, jointly optimizing over all parameter equations is computationally infeasible because of the scale of the resulting problem. These observations motivated our block-wise optimization. To further evaluate the importance of the block construction, we have now added an ablation comparing two strategies. In our original method, parameters are assigned to blocks sequentially, such that parameters from the same layer, or from nearby layers, tend to be grouped together. In the alternative variant, parameters are assigned randomly to blocks. All other components of the method are kept unchanged.
>
> | Block construction | AUC $\uparrow$ | TPR @ 0.01% FPR $\uparrow$ | TPR @ 0% FPR $\uparrow$ |
> |:---|---:|---:|---:|
> | Random Blocks | 0.80 | 0.37% | 0.08% |
> | Sequential Blocks | **0.81** | **2.76%** | **1.41%** |
>
> Sequential block construction performs better, particularly in the low-FPR regime. We attribute this improvement to the more homogeneous statistics within each block: sequential blocks tend to contain parameters from the same or nearby layers, whereas random blocks mix equations associated with parameters that may have substantially different scales and distributions. For example, the row-norm coefficient of variation is (0.48) for sequential blocks versus (0.91) for random blocks, and the mean absolute row correlation is (0.163) versus (0.069). Thus, together with the ablation above, the results suggest that sequential blocks improve the attack by grouping parameters with similar gradient statistics, leading to a more stable and effective coefficient-recovery procedure. Although this experiment does not directly measure the condition number of the optimization problem, it provides empirical evidence that grouping equations with similar statistics results in a more stable and effective optimization procedure.  We have clarified this claim and added the new ablation to the revised manuscript (see App. B.7).
>
> ---
>
> **2. Justification of the precise training configuration change for black-box attacks: How can we decouple the "knowledge of training configuration" from the "strength of the reference model" to more strongly support the claim?**
>
> **Response:**
> To ensure a fair comparison, we selected training configurations for both the target and reference models that yield good convergence and strong classification accuracy. To demonstrate that the performance degradation of the reference-model-based attacks is not simply due to weaker or undertrained reference models, we now compare the training and test accuracies of models trained on CIFAR-10 across two configurations: the standard configuration used in our main experiments and the configuration used in the "Unknown Training Configuration" setting. We have also added this comparison to the revised manuscript (see App. B.4).
>
> | Training configuration | Average train accuracy | Average test accuracy |
> |:---|---:|---:|
> | Standard configuration | 100.0% | 89.9% |
> | Unknown Training Configuration | 99.9% | 88.0% |
>
> As shown, both configurations achieve nearly perfect training accuracy and strong test accuracy, with only a small difference in classification performance. This indicates that the reference models in the unknown-configuration setting are well trained and have converged sufficiently.

---

> ### Author Response · Authors · 2026-06-12
> **Official Comment by Authors**
>
> **3. It would be helpful to try this attack on one different architecture to see if these claims are specific to this ResNet model.**
>
> **Response:**
> Table 2 reports results for both VGG16 and ResNet50, in addition to our main results on ResNet18, showing that the attack is not specific to a single ResNet architecture. Prior implicit-bias work has mainly studied MLPs and relatively simple networks, and the theory applies directly only to homogeneous models. Our results provide empirical evidence that the proposed signal remains effective for more complex architectures, including VGG16 and ResNet50.
>
> ---
>
> **Broader Impact Concerns: A Broader Impact section should be added to list applications where these may be used for malicious purposes.**
>
> **Response:**
> We agree and have added a Broader Impact Concerns section to the revised paper (see Section F in the appendix).